# Biochemical and Functional Characterization of Kidney Bean Protein Alcalase-Hydrolysates and Their Preservative Action on Stored Chicken Meat

**DOI:** 10.3390/molecules26154690

**Published:** 2021-08-03

**Authors:** Ahmed M. Saad, Mahmoud Z. Sitohy, Alshaymaa I. Ahmed, Nourhan A. Rabie, Shimaa A. Amin, Salama M. Aboelenin, Mohamed M. Soliman, Mohamed T. El-Saadony

**Affiliations:** 1Biochemistry Department, Faculty of Agriculture, Zagazig University, Zagazig 44511, Egypt; AMMousttafa@agri.zu.edu.eg; 2Department of Agricultural Microbiology, Faculty of Agriculture, Beni-Suef University, Beni-Suef 62511, Egypt; Alshaymaaibrahim@gmail.com; 3Department of Food Science, Faculty of Agriculture, Zagazig University, Zagazig 44511, Egypt; Rabienourhan882@gmail.com; 4Department of Agricultural Microbiology, Faculty of Agriculture, Ain-Shams University, Cairo 11566, Egypt; shimaa_amin@agr.asu.edu.eg; 5Biology Department, Turabah University College, Taif University, P.O. Box 1109, Taif 21944, Saudi Arabia; s.aboelenin@tu.edu.sa; 6Clinical Laboratory Sciences Department, Turabah University College, Taif University, P.O. Box 11099, Taif 21944, Saudi Arabia; mmsoliman@tu.edu.sa; 7Department of Agricultural Microbiology, Faculty of Agriculture, Zagazig University, Zagazig 44511, Egypt; m.talaatelsadony@gmail.com

**Keywords:** legume protein isolation, enzymatic hydrolysis, phaseolin, antioxidant, antimicrobial, chicken meat cold storage

## Abstract

A new preservation approach is presented in this article to prolong the lifetime of raw chicken meat and enhance its quality at 4 °C via coating with highly soluble kidney bean protein hydrolysate. The hydrolysates of the black, red, and white kidney protein (BKH, RKH, and WKH) were obtained after 30 min enzymatic hydrolysis with Alcalase (E/S ratio of 1:100, hydrolysis degree 25–29%). The different phaseolin subunits (8S) appeared in SDS-PAGE in 35–45 kD molecular weight range while vicilin appeared in the molecular weight range of 55–75 kD. The kidney bean protein hydrolysates have considerable antioxidant activity as evidenced by the DPPH-scavenging activity and β-carotine-linolenic assay, as well as antimicrobial activity evaluated by disc diffusion assay. BKH followed by RKH (800 µg/mL) significantly (*p* ≤ 0.05) scavenged 95, 91% of DPPH and inhibited 82–88% of linoleic oxidation. The three studied hydrolysates significantly inhibited the growth of bacteria, yeast, and fungi, where BKH was the most performing. Kidney bean protein hydrolysates could shield the chicken meat because of their amphoteric nature and many functional properties (water and oil-absorbing capacity and foaming stability). The quality of chicken meat was assessed by tracing the fluctuations in the chemical parameters (pH, met-myoglobin, lipid oxidation, and TVBN), bacterial load (total bacterial count, and psychrophilic count), color parameters and sensorial traits during cold preservation (4 °C). The hydrolysates (800 µg/g) significantly *p* ≤ 0.05 reduced the increment in meat pH and TVBN values, inhibited 59–70% of lipid oxidation as compared to control during 30 days of cold storage via eliminating 50% of bacterial load and maintained secured storage for 30 days. RKH and WKH significantly (*p* ≤ 0.05) enhanced *L**, *a** values, thus augmented the meat whiteness and redness, while, BKH increased *b** values, declining all color parameters during meat storage. RKH and WKH (800 µg/g) (*p* ≤ 0.05) maintained 50–71% and 69–75% of meat color and odor, respectively, increased the meat juiciness after 30 days of cold storage. BKH, RKH and WKH can be safely incorporated into novel foods.

## 1. Introduction

Poultry meat is one of the most popular foods worldwide. However, increasing consumer awareness about meat origin, animal welfare, and meat safety challenges the production of high-quality meat. Microbial contamination, lipid, and protein oxidation are the main problems affecting chicken meat quality. Consumer rejection and economic losses might result from muscle components oxidation and discoloration, as well as the contamination of chicken meat with foodborne pathogens during processing and marketing or the inappropriate meat cooking, cooling or storage causing illness or sometimes death. These factors cost billions of dollars in medical and social care [1]. Centers for Disease Control and Prevention (CDC) estimates that foodborne pathogens lead to 48 million sick people, 128,000 hospitalized people, and 3000 deaths every year [2]. Currently, consumers need high-quality food with extended shelf life and reasonable prices. Therefore, food producers employ researchers to find food additives that meet consumer desires. Chemical additives, i.e., nitrate, butylated hydroxyanisole (BHA), and butylated hydroxytoluene (BHT) are highly efficient in preserving foods and prolonging the storage period by reducing chemical and biological changes [3,4]. However, these chemicals are expensive and may cause various damages to health when intensively used. Therefore, consumers prefer natural preservatives. Researchers sought to preserve various foods with biologically active natural additives [5,6,7,8,9]. The new trend in food preservation uses bioactive peptides as natural additives to preserve milk [10,11,12,13,14,15], meat products [16,17,18]. These proteins can generate bioactive peptides under enzymatic hydrolysis [19,20,21]. Fermentation and digestion in the gastrointestinal (GI) can also generate bioactive peptides [22]. Bioactive peptides consist of about 2–20 amino acids and have a relatively low molecular weight compared to proteins [23], facilitating their absorption into the small intestine and promoting their biological effects [24]. The behavior of these bioactive peptides depends on their weight and length, their structure and sequence of amino acids, hydrophobic/hydrophilic properties, spatial structure, and charge character of the constituting amino acid [25,26]. Biologically active peptides, especially isolated from plants are characterized by their antioxidant and antimicrobial activities. Therefore, they have a role in improving the food quality and technological properties. [27].

White, red, and dark kidney beans (*Phaseolus vulgarus* L.) members of *Fabaceae*, have a considerable content of protein (25–44%), and other nutrients [28,29]. Various studies investigated that common beans contributed in many diseases treatments, i.e., cancer and diabetes [30]. Phaseolin “8S” is the main storage protein in kidney bean [31]. Phaseolins have medium molecular weight in the range 43–55 kD and are encompassing potentially bioactive oligopeptides of 2–20 amino acids’ sizes [32]. These bioactive peptides (BPs) inherent in the legume seeds can be liberated and produced under the action of different proteases such as Alcalase, Pepsin, Trypsin, and Papain [21]. These BPs were reported to exhibit antibacterial activity against various spoilage bacteria and exert antioxidant activity against lipid oxidation [9,15,18]. Therefore, they can be incorporated into different functional foods [33]. Alcalase is endopeptidase enzyme isolated from *Bacillus licheniformis*, it can hydrolyze several proteins to small peptides. The hydrolysis activity of alcalase was more activity than trypsin and papain [34]. Enzymatic hydrolysis reduces the polypeptide size, and increase the bioavailability of amino acid in the human and animal diet [35]. Bumrungsart and Duangmal [36] explored the influence of Flavourzyme^®^ (1–7%) on a two h hydrolysis of gram bean protein isolate on the structure and antioxidant activity of the resulting peptides [37]. Saad, et al. [21] investigated the influence of incorporating various protein hydrolysates into meat products and juices as potential natural preservatives [38,39]. In this study, three protein hydrolysates (WKH, RKH, and BKH) were obtained from white, red, and black kidney bean protein by 30-min Alcalase hydrolysis. The hydrolysates were physicochemically characterized. The preservative action of hydrolysates were followed on stored chicken meat at cold conditions for 30 days.

## 2. Materials and Methods

Black, red, and white kidney bean seeds (*Phaseolus vulgaris* L) were acquired from a local market in Cairo City, Egypt, and raw chicken meat samples were obtained from a private farm at Zagazig City, Egypt. Alcalase enzyme and DPPH were from (Sigma-Aldrich, Inc., St. Louis, MO 68178, USA), and Muller Hinton agar (Oxoid, Basingstoke, UK). Electrophoresis reagents were from Bio-Rad laboratories (Sigma-Aldrich, Inc., St. Louis, MO 68178, USA). All chemicals were of analytical grade. The microorganisms used in this study to fully assess the antimicrobial activity included; *Bacillus cereus*, *Listeria monocytogenes*, *Staphylococcus pyogenes*, *Escherichia coli*, *Campylobacter jejuni*, *Salmonella typhi*, *Aspergillus niger*, *Aspergillus fumigatus*, *Aspergillus flavus*, *Penicillium chrysogenum*, *Fusarium exquisite*, *Fusarium avenaceum*, *Candida gelbeta*, *Candida tropicalis*, *Candida albicans*, *Rhodotorula rubra*, *Rhodotorula minuta*, and *Rhodotorula mucilginosa.* These isolates were obtained from Agricultural Microbiology Department, Faculty of Agriculture, Zagazig University, Zagazig, Egypt.

### 2.1. Protein and Peptides Isolation

#### 2.1.1. Black, Red and White Kidney Bean Protein Isolates (BKI, RKI, and WKI)

The black, red, and white bean seeds were cleaned and ground by Moulinex blender Type 716 (France) to pass through a 1 mm 2 sieve. The powder was defatted by hexane in a Soxhlet apparatus. WKI, RKI, and BKI were isolated from defatted powder according to the method of [40], with some modifications. Defatted flours were homogenized in distilled water (5% *w*/*v*), and pH was adjusted to 8 with NaOH (2 N). The suspension was stirred for two hours at room temperature before centrifugation at 4000× *g*, 30 min. The pH of supernatants was adjusted to 4.5 with HCl (2 N), and the resulting precipitates (proteins) were recovered by centrifugation at 4000× *g*, 30 min, homogenized again in distilled water, and pH was adjusted to 7.0 with 2 N NaOH, dialyzed, then lyophilized.

#### 2.1.2. Black, Red and White Kidney Bean Alcalase Enzymatic Hydrolysates

The BKI, RKI, and WKI were blended with Alcalase (1:100, *w*/*w*) then dissolved in phosphate buffer pH 6, and incubated at 37 °C for intervals (0, 30, and 60 min). The temperature was raised to 90 °C for 15 min to inhibit the enzyme. The White, red, black kidney bean hydrolysates (BKH, RKH, and WKH) were obtained by centrifugation at (4000× *g*, 30 min), lyophilized and kept at −20 °C for further analysis [15,21].

### 2.2. Biochemical Characterization of White, Red, Black Kidney Bean Protein and Hydrolysates

#### 2.2.1. Degree of Hydrolysis (DH)

The DHs of BKI, RKI, and WKI after 30 and 60 min were estimated by the method of Holye and Merrltt [41]. One mL of protein isolate supernatant was dissolved in trichloroacetic acid (TCA, 10%) (1:1 *v*/*v*), then centrifuged at (10,000 rpm, 10 min, 4 °C) (Sigma 3–30 k, Germany) to collect TCA supernatant. Total nitrogen in TCA supernatant and the protein isolates was measured by the Kjeldahl method [21]:(1)$$\mathrm{Degree}\;\mathrm{of}\;\mathrm{hyrolysis}\;\left(\%\right)=\frac{\mathrm{TCA}\;10\%-\mathrm{Soluble}\;\mathrm{nitrogen}\;\mathrm{in}\;\mathrm{the}\;\mathrm{sample}}{\mathrm{Total}\;\mathrm{nitrogen}\;\mathrm{in}\;\mathrm{the}\;\mathrm{sample}}\times 100$$

#### 2.2.2. Sodium Dodecyl Sulfate Polyacrylamide Gel Electrophoresis (SDS-PAGE)

The BKI, RKI, and WKI and alcalase hydrolysates (WKH, RKH, and BKH) were fractionated by discontinuous SDS-PAGE according to Laemmli [42]. An amount of 20 µg of peptides was dispersed in 100 μL of reducing SDS-loading sample buffer pH 6.8, heated at 96 °C for 3 min then was centrifuged at 14,000 rpm for 10 min and 5 μL aliquot was electrophoresed (5 μL of protein/lane). In the present procedure, the resolving (18%) and stacking gels (10%) were prepared. The electrophoresis was run toward the positive pole at 10 mA on the stacking gel and 20 mA on the resolving gel. After electrophoretic separation, the gels were carefully removed and the bands was stabilized in 300 mL of 10% trichloroacetic acid (TCA) for 12 h at room temperature. The protein bands were developed with Commassie Brilliant Blue R-250 dye and molecular weight of separated proteins bands were identified by ChemiDoc Gel documentation (BioRad, Hercules, CA, USA). A molecular weight marker in the range of 6.5 to 245 kDa was used.

#### 2.2.3. Chemical Analysis of Hydrolysates

The proximate analysis (Protein, Carbohydrates, Fat, Ash, and Moisture) of WKH, RKH, and BKH were estimated according to Roy, et al. [43].

#### 2.2.4. Functional Properties of Hydrolysates

##### Hydrolysates Solubility

The solubility of protein hydrolysates was measured at pH range (2–10) according to [44] with slight modifications. An amount of 100 mg of protein hydrolysates was homogenized in 25 mL of distilled water and pH was adjusted using pH meter (pH 211 HANNA instruments Inc. Woonsocket USA made in Romania). The protein hydrolysates suspensions were stirred for 1 h at 30 °C, and then centrifuged at (1200× *g*, 20 min, 4 °C) to obtain the supernatant then protein content was measured by the Kjeldahl method [45]. The peptides solubility (%) was measured against pH following the equation:(2)$$\mathrm{Solubility}\;\left(\%\right)=\frac{\mathrm{Protein}\;\mathrm{content}\;\mathrm{in}\;\mathrm{supernatant}}{\mathrm{Protein}\;\mathrm{content}\;\mathrm{in}\;\mathrm{sample}}\times 100$$

##### Water Absorption Capacity

Water absorption capacity was estimated as Wani, et al. [46] with slight modifications; 1 g of each protein hydrolysate was mixed with 10 mL of distilled water in weighted test tubes, stirred for 30 min, then centrifuged at 6000× *g* for 30 min, and the supernatant was discarded. The tubes remained at 45 °C for 25 min at a 45° angle until surface water was removed, then weighted again. Water absorption percentage was expressed as g of absorbed water/g of sample.

##### Oil Absorption Capacity

Oil absorption capacity was determined as Wani, et al. [46] with some modifications. 0.5 g of each hydrolysate was added to 6 mL of oil in weighted test tubes, stirred for 30 min, then centrifuged at 6000× *g* for 30 min, and the supernatant was discarded. After removing the supernatant, the tubes were kept upside-down for 25 min to remove the excess oil and weighted again. Oil absorption percentage was calculated as g absorbed oil/g of sample.

##### Foaming Capacity and Foaming Stability

Foaming capacity was estimated according to Wani, et al. [47] with slight modifications; 1 g of each hydrolysate was mixed at high speed with 50 mL distilled water in a Moulinex mixer Type 716 (France) for 5 min. Then, the solution was poured into a scaled cylinder (250 mL) and foam volume was read after 30 s. The foam stability was determined by recording the foam volume through time. Foaming activity was calculated using the following equation:(3)$$\mathrm{FC}\;\left(\%\right)=\frac{\left(A0\;–\;B\right)\times 100\;}{B}\;$$
where % foaming capacity (FC), is, and sample volume before stirring (B), and sample volume after stirring (A0).

#### 2.2.5. Chemical Constituents of the Hydrolysates

##### Total Phenolic Compounds (TPC) in The Hydrolysates

Polyphenols were assessed in BKH, RKH, and WKH suspension and calculated as mg GAE/g following Folin-Ciocalteu method Škerget, et al. [48], the absorbance was examined at 750 nm using a spectrophotometer (JENWAY 6405 UV/visible, UK) and was applied in standard Gallic acid linear (equation: y = 0.004x + 0.1257).

##### Total Flavonoids in the Hydrolysates

An aliquot (100 µL) of each hydrolysate suspension was homogenized in 3 mL of ethanoic AlCl_3_ and kept for an hour in the dark as per [49]. The absorbance at 430 nm was measured and applied in the standard quercetin linear (equation: y = 0.0025x − 0.0057), to obtain total flavonoids content as µg QE/mL.

##### Total Anthocyanin Content (TAC) in BKH, RKH, and WKH Peptides

Total anthocyanin content in BKH, RKH, and WKH was evaluated at different pH (1 and 4.5) by spectrophotometer [50]. The change in absorbance of the supernatant was measured at 520 and 700 nm. The total monomeric anthocyanin was presented as cyanidin-3-Glu equivalent (mg/g) in the following Equation (4):(4)$$\mathrm{Amount}\;\mathrm{of}\;\mathrm{anthocyanin}\;\left(\mathrm{mg}/g\right)=\frac{A\times 449.2\times \mathrm{DF}\times 10^{3}}{26,900}\;A=\left(A520\;–\;A700\right)\mathrm{pH}\;1.0\;–\;\left(A520\;–\;A700\right)\mathrm{pH}\;4.5\;$$

#### 2.2.6. Biological Activity of Hydrolysates

##### Antioxidant Activity

DPPH Assay

The radical scavenging activity of BKH, RKH, and WKH levels (50, 100, 200, 400, and 800 µg/mL) was deduced from the ability to convert the purple color of DPPH˙ to yellow as compared to TBHQ as synthetic antioxidant according to Hatano, et al. [51] with some modifications. An aliquot (100 µL) of each hydrolysate level was mixed with 1 mL ethanolic DPPH and incubated for 30 min in the dark. The absorbance (Abs) was estimated at 517 nm against the control. DPPH˙ antiradical activity (%) was calculated in the following equation:(5)$$\mathrm{Radical}\;\mathrm{scavenging}\;\mathrm{activity}\;\left(\%\right)=\frac{\left(\mathrm{Abs}.\mathrm{control}-\mathrm{Abs}.\mathrm{sample}\right)}{\left(\mathrm{Abs}.\mathrm{control}\right)}\times 100$$

β-Carotene/Linoleic Acid Bleaching

The ability of BKH, RKH, and WKH (50, 100, 200, 400, and 800 µg/mL) to prevent discoloration of β-carotene was determined following Dastmalchi, et al. [52]. In brief, 0.2 mg of β-carotene was homogenized in 1 mL of chloroform, 20 mg of linoleic acid, and 200 mg of tween 20 were added in a round-bottom flask. The chloroform was evaporated, and 50 mL of distilled water was added, and the mixture was vigorously stirred. An aliquot (0.5 mL) of each hydrolysate suspension was mixed with 3 mL of the β-Carotene/linoleic emulsion, and the absorbance (Abs0) was measured at 470 nm was recorded. The other samples were incubated at 50 °C for 120 min, and the absorbance was similarly recorded at 470 nm (Abs120). A control without hydrolysate was similarly assayed. Antioxidant activity was calculated as follows:(6)$$\mathrm{Antioxidant}\;\mathrm{activity}\;\left(\%\right)=1-\frac{\mathrm{Abs}0\;\mathrm{sample}\;-\;\mathrm{Abs}120\;\mathrm{sample}}{\mathrm{Abs}0\;\mathrm{control}\;-\;\mathrm{Abs}120\;\mathrm{control}}\times 100\;$$
where Abs_0_ sample is the absorbance of sample at 0-time, Abs_120_ sample is the absorbance after 120 min, Abs_0_ control is the absorbance of control at 0-time, and Abs_120_ control is the absorbance of control after 120 min.

##### Antimicrobial Activity

Antibacterial

Antibacterial activity was estimated by disc diffusion assay according to El-Saadony, et al. [53]. Muller Hinton agar (MHA) plates were inoculated with activated bacterial isolates. Discs (6 mm) were saturated with protein hydrolysates concentrations (50, 100, 200, 400, and 800 µg/mL) then placed on the surface of MHA plates and incubated for 24 h at 37 °C. The discs were saturated with sterilized distilled water used a control. The ruler was used to measure the zones of inhibition surrounding the discs (mm). The least concentration-inducing growth inhibition of bacteriaa was considered the minimum inhibitory concentration (MIC). The MIC was estimated by adding 50 μL of each level of tested hydrolysates (50, 100, 200, 400, and 800 µg/mL) to 9 mL MHB inoculated with 100 μL pathogenic bacteria. The tubes were incubated for 24 h at 37 °C, and the turbidity was recorded at 600 nm [54]. The least concentration totally kill the tested bacteria was, the minimum bactericidal concentration (MBC) was estimated by spreading 100 μL of each hydrolysate MIC on MHA plates’ surface then incubated at 37 °C for 24 h and the bacterial growth was observed.

Antifungal

The tested *Candida* and fungi species were cultured on Sabouraud dextrose agar (SDA) plates, and were incubated at 37° and 30 °C for 2, and 5 days, respectively. The antifungal activities of BKH, RKH, and WKH (50, 100, 200, 400, and 800 µg/mL) were evaluated using the disc assay [15]. The SD broth tubes inoculated with *Candida* spp., or fungal spore suspension were incubated at 37 °C and 30 °C for 2 and 5 day, respectively to get inoculum (10^5^ CFU ml^−1^) concentration. An aliquot (100 µL) of *Candida* and fungal inoculum was spread over SDA plates. The saturated paper discs (6 mm) with BKH, RKH, and WKH (50, 100, 200, 400, and 800 µg/mL) were placed on the sides of cultivated SDA plates. The plates were incubated at 37 °C for 2 days (*Candida*) and 30 °C for 5 days (Fungi). The resultant zones around the discs were measured by ruler indicating antifungal activity of tested hydrolysates. The minimum inhibitory concentration (MIC) of the hydrolysate was evaluated by the micro-dilution broth method [55]. The SDB tubes contained hydrolysates concentrations and *Candida* or fungal inoculum were incubated at 37 °C for 2 day (*Candida*) or 30 °C for 5 days (fungi). The obtained turbidity was read at 600 nm using a spectrophotometer (Shimadzu Corporation, Analytical Instruments Division, Kyoto, Japan). The lowest hydrolysate concentration inhibiting the fungal growth, was recorded as the MIC and the lowest concentration totally removing the fungal growth was considered the minimum fungicidal concentration (MFC) [55]. The MIC levels of BKH, RKH, and WKH was sub-culturing the onto SDA plates to calculate minimum fungicidal concentration (MFC). The plates were incubated at 30 °C for 3–5 days (fungi) or 37 °C for 2 day (*Candida*).

### 2.3. Preservation of Raw Chicken Meat

Chicken muscles (500 g) were cut with a sterile knife to 4 cm-cubes. The meat samples were transferred to sterilized polyethylene packets, then divided into 4 groups (control; T1, chicken meat cubes soaked in different concentrations of WKH (200, 400, and 800) for 24 h; T2, chicken meat cubes soaked in different concentrations of RKH, and T3, chicken meat cubes soaked in different concentrations of BKH). All samples were air-dried under sterilization conditions. The treated samples were stored in plastic bags and stored at 4 °C for 4 weeks for different analysis.

#### 2.3.1. Physicochemical Analysis of Meat Sample

##### pH Estimation

Five grams of meat samples were minced in a sterilized mincer and homogenized in 45 mL distilled water for 30 min, and filtrated. The pH of chicken meat filtrate pH was estimated as Özyurt, et al. [56] using a pH-meter (pH 211 HANNA instruments Inc., Woonsocket, RI, USA).

##### Metmyoglobin (MetMb) Analysis

The MetMb contents in raw chicken meat samples were determined as per Badr [57]. Chicken meat samples (5 g) were homogenized in 25 mL of 40 mM phosphate buffer pH 6.8, stirred for 5 min at 4 °C, and kept for 1 h at 4 °C before centrifugation at (500× *g*, 30 min, 4 °C. The supernatant absorbance was measured using a spectrophotometer (JENWAY 6405 UV/visible, UK) at wavelengths of 525, 545, 565, and 572 nm. MetMb (%) was calculated using the following equation according to Krzywicki [58]:(7)$$\mathrm{MetMb}\;\left(\%\right)=\left[-2.51\left(A572|A525\right)+0.8\left(A545|A525\right)+0.777\left(A565|A525\right)\right]\times 100$$

##### Lipid Peroxidation Assay

Lipid peroxidation inhibition (%) was determined using Niehaus Jr and Samuelsson [59] procedure. Chicken meat filtrate was dissolved in phosphate buffer (50 mM, pH 7) and centrifuged at (12,000× *g*, 60 min, 4 °C) to obtain the supernatant. 100 µL of the supernatant was added to 2 mL of TBA, boiled for 30 min, then cooled. The absorbance was estimated at 535 nm using a spectrophotometer (JENWAY 6405 UV/visible, UK) as compared to control. The inhibition in lipid oxidation (%) was calculated using the following equation:(8)$$\mathrm{Lipid}\;\mathrm{oxidation}\;\mathrm{inhibtion}\;\left(\%\right)=1-\frac{\mathrm{Sample}\;\mathrm{absorbance}}{\mathrm{control}\;\mathrm{absorbance}}\times 100$$

##### Total Volatile Basic Nitrogen (TVBN)

Ten milliliters of chicken meat filtrate was combined with 30 mL of 5% Trichloroacetic acid and centrifuged at 3000× *g* for 1 h. Five milliliters of the supernatant was pipetted into the Markham apparatus, and 5 mL of NaOH (2 N) was added. The obtained steam was titrated into 15 mL of standard HCl 0.01 N containing drops of rosolic indicator. The excess acid was then titrated in the receiving flask containing phenolphthalein with standard NaOH (0.01 N) to a pale pink end. A technical blank was done using 5 mL Trichloroacetic acid without sample and titrated as before Idakwo, et al. [60]. The concentration of TVBN (mg N/100 g sample) was calculated as follows:(9)$$\mathrm{TVBN}\;\left(\mathrm{mg}\frac{N}{100}g\;\mathrm{sample}\right)=\frac{\left(N\right)\left(\mathrm{VB}\;-\;\mathrm{VS}\right)\left(14\right)\left(300+\;W\right)}{5}\;$$
where VB = the amount of NaOH (mL) in blank titration, W = mositure content of sample in g/100 g, N = Standard NaOH Normality, and vs. = the amount of NaOH (mL) in the main titration.

#### 2.3.2. Sensory Evaluation and Color Measurement

The shearing force of cooked chicken meat cubes enriched with BKH, RKH, and WKH at different levels (200, 400, and 800 µg/g) was measured by a texture analyzer (Compac-100 model, Sun Scientific Co., Tokyo, Japan) using the method of Piga, et al. [61]. The meat samples were coded with three random digits, and were served to evaluation by eight experienced panelists at Zagazig University, Egypt. Each panelist evaluated five attributes; tenderness, juiciness, taste, aroma, and overall acceptability in each sample at interval of (0, 10, 20, and 30) days of cold storage by using a 9-point hedonic scale (9 = like extremely, and 1 = dislike extremely). Samples with scores below 5 were deemed unacceptable.

The color parameters (L*, a* and b*) of chicken meat color were evaluated by a color analyzer (Hunter Lab color Flex EZ, USA). The L* expressed [(0) whiteness to (100) darkness], a* expressed [(−) redness to (+) greenness] and b* value observed [yellowness (+) to blueness (−)] and total color change (∆E) was calculated from Equation (10). Chicken meat samples were placed into analyzer flask, and then the colorimeter sensor measured color attributes [62]:(10)$$∆E=\sqrt{{\left(L_{2}-L_{1}\right)}^{2}+{\left(a_{2}-a_{1}\right)}^{2}}+{\left(b_{2}-b_{1}\right)}^{2}$$

#### 2.3.3. Microbial Analysis

The microbial load of raw chicken meat was estimated according to Saad, et al. [63]. The meat filtrate was dissolved in sterilized peptone buffer (1:9 *w*/*v*) in a screw bottle and stirred for 10 min to prepare 10^−1^ concentration. One milliliter of the suspension was used in preparing decimal dilutions down to 10^−5^. One mL of each dilution was added in one-use Petri-dishes, then the specific media was added [64]. After 24 h incubation at 30 °C, the total bacterial count (TBC) was enumerated on plate count agar (PCA). Additionally, PCA was used for counting psychrophilic bacteria count (PBC) after incubating for 10 days at 7 °C. Microbiological results were converted to logarithms (CFU/g) [65,66,67].

### 2.4. Statistical Analysis

All tests were carried out in triplicate. The means of the experimental data triplicates were subjected to the analysis of variance (ANOVA) test with a significance level of 5%, followed by LSD test to investigate the significant differences using Data analysis in Microsoft Excel with Microsoft Office version 2019.

## 3. Results and Discussion

### 3.1. Physico-Chemical Characterization of the Hydrolysates

#### 3.1.1. Chemical Composition of Protein Isolates and Hydrolysates

The proximate compositions of white, red, and black kidney bean plants, protein isolates, and hydrolysates are shown in Table 1. Black kidney bean has higher protein contents in the different forms, total, isolate and hydrolysates (T, I, and H), amounting to 24.5, 91, and 88%, respectively. Alternatively, red kidney beans had the highest carbohydrate content in the different forms (T, I and H), recording 65.1, 35.4, and 8.7%, respectively. Ash content was, however, highest in all forms of white kidney bean (T, I, and H), reaching 5.1, 7.2, and 9.2%, respectively. Evangelho, et al. [68] found that black bean proximate composition was 23.1% protein, 4.3% ash, 1.2% fat, 71.4% of total carbohydrate, and protein isolate was 81.6% protein, and 11.4% carbohydrates. Generally, ash content increased with protein isolation or hydrolysis by Alcalase, but protein content significantly increased by 73–75% in isolate forms (BKI, RKI, and WKI), and slightly decreased in the hydrolysates forms (BKH, RKH, and WKH), probably due to the increments in ash content by 11–21%. A similar reduction in protein content of black bean protein was observed in a previous study after 2 h-Alcalase hydrolysis. These reductions were associated with an increase in the ash content of black bean protein hydrolysates, possibly as a result of supplying inorganic alkali to keep the pH constant during hydrolysis [68]. A similar reduction in protein content accompanied by an increase in ash content was observed in Alcalase-hydrolyzed chickpea protein [69].

#### 3.1.2. SDS-PAGE Electrophoretic Pattern

SDS-PAGE electropherogram of BKI, RKI, and WKI showed in lanes 1, 2, and 3, respectively (Figure 1). Seventeen bands were detected in lanes (1–3) with molecular weights in the range 11–135 kD, referring to similar protein patterns in the three cultivars of kidney beans. Storage proteins, i.e., vicilin (7S), and phaseolin (8S), were found within the molecular weights between 40–75 kD. The bands representing phaseolin (40–48 kD), seem more intense than those representing vicilin in accordance with Los, et al. [32], indicating that phaseolin was the most abundant protein of the common bean. These bands were relatively the least affected by the 30-min Alcalase hydrolysis as manifested in the electrophoretic pattern of BKH, RKH, and WKH in lanes 4, 5, and 6, respectively. Extending the Alcalase hydrolysis time to 60 min led to the disappearance of these bands (data not shown). The observed resistance of phaseolin to enzymatic hydrolysis in the opposite to vicilin, the total disappearance of bands representing 7S (vicilin) but not phaseolin bands, which relatively resisted enzymatic hydrolysis. Likewise, the 42–47 kD bands referring to phaseolin were visualized in SDS-PAGE profiles of black bean protein isolate [70] after 2 h pepsin hydrolysis while higher molecular weight bands (>50 kD) disappeared. Similar resistance of phaseolin was noticed after six h pepsin hydrolysis of white kidney protein isolate [21]. The observed higher intensity of phaseolin bands in BKH and RKH than WKH might have originated from the higher phenolic content in black and kidney beans, as de Toledo, et al. [71] mentioned.

#### 3.1.3. Degree of Hydrolysis (DH)

The data in Figure 2A show the hydrolysis degree of black, red, and white kidney bean protein isolates after subjection to Alcalase for 60 min at 37 C. The DH was significantly (*p* ≤ 0.05) increased with hydrolysis time but not significant differences between DH values after 30 and 60 min. The highest DH of BKH, RKH, and WKH were obtained after 30 min, reaching 29, 27, and 25%. So, the BKH exhibited the highest DH with about a 14% relative increase over WKH and RKH. Similar results were obtained by Saad, et al. [21] who found that DH of kidney bean protein isolate was 33% after hydrolysis with pepsin 1% for 6 h. In addition, Abdel-Hamid, et al. [72] found that camel whey protein isolates was hydrolyzed by 27% by Alcalase after 4 h. Furthermore, do Evangelho, et al. [70] found the highest DH = 27% of pepsin black bean protein hydrolysate achieved after 120 min. Moreover, the highest DH (75%) when Bumrungsart and Duangmal [36] used Flavourzyme^®^ (6%, 360 min) to hydrolyze black gram bean protein isolate. The obtained DH in the range of previous studies.

#### 3.1.4. pH-Protein Solubility

The inclusion of functional proteins in food formulations depends on its solubility, which affects other protein functional properties, such as foaming and emulsifying [73]. The isoelectric pH (lowest solubility) of WKH, RKH, and BKH ranged 4–6 (Figure 2B), which is not different from the intact mother protein. Generally, the solubility of protein significantly increased *p* ≤ 0.05 when the pH moves away from the isoelectric point. The solubility of WKH, RKH, and BKH after 30 min hydrolysis by Alcalase was (53, 66, and 75%) at pH 2 and recorded the highest levels at pH 10, i.e., 70, 75, and 90%, respectively. The order of solubility levels BKH > RKH> WKH, is a reflection of the same order of the degree of hydrolysis, which understandably enhances it. The observed higher solubility at the basic side than the acidic side agrees with Los, et al. [32] who reported 3 5.13% solubility values of carioca bean and soybean protein hydrolysate at pH 3.0 against 100% at pH 7.0. Osman, et al. [74] found that white kidney bean protein pepsin hydrolysate has maximum solubility at pH 10 with 82%, also, Wahdan and Saad [75] found that white kidney bean protein papain hydrolysate has 78% solubility at pH 10.

#### 3.1.5. Functional Properties

The data in Table 2 show the functional properties of the hydrolysates; BKH, RKH and WKH, after 30 min Alcalase-hydrolysis. The BKH has the highest water holding capacity (WHC), recording about a 30% solubility increase over the original intact proteins (BKPI). The other two hydrolysates, RKH and WKH, recoded lower relative solubility increases over their intact protein (RKI and WKI), amounting to 15%, and 20%, respectively. The oil absorbance capacity (OAC) of BKH recorded the highest significant (*p* < 0.05) increase of 35% over the intact protein (BKI), against only 20% and 25% relative increases in the case of RKH, and WKH over their intact mother forms. On the other hand, the three hydrolysates showed higher foaming stability at pH 5, i.e., 88, 75, and 50 min for BKH, RKH, and WKH, respectively, with relative increases about 25–40% over the respective intact protein forms, i.e., BKI, RKI and WKI. Similar results were reported by Wani, et al. [47], who found higher forming stability of black gram protein hydrolysate at pH 5 than pH 3 and 7. The foaming stability index (FSI) of protein hydrolysates was in the range of 15.7–89.0 min. Generally, protein hydrolysates have higher ESI at pH 3 and 5, against lower pH 7. In addition, Eckert, et al. [76] stated that foaming capacity (FC) of fava bean protein pepsin-hydrolysate increased by 74% at pH 5 and 50% at pH 7 over the intact protein. In addition, the oil holding capacity (OHC) increased from 6.12 to 8.21% g/g by pepsin hydrolysis. Generally, the enzymatic hydrolysis significantly increased the functional properties of protein isolates. Faustino, et al. [27] stated that the inclusion of high solubility proteins in foods enhances the technological properties in supplemented foods, and it was required in many functional food applications.

#### 3.1.6. Total Phenolic, Flavonoids, and Anthocyandins Content

Table 3 presents the contents of total phenolic, flavonoids, and anthocyanins in the different protein hydrolysates. Generally, the contents of these compounds significantly increased in a concentration-dependent manner. The BKH (800 µg/mL) recorded higher values of phenolic, flavonoids, and anthocyanins amounting to 63.3 mg GAE/g, 16.2 mg QE/g, and 0.6 mg/mL (800 µg/mL), respectively. The levels of phenolic and flavonoids were not significantly different among the three protein hydrolysates (BKH, RKH and WKH) but the levels of anthocyanin in RKH and WKH were significantly (*p* ≤ 0.05) lower than in BKH, recording 91, and 45 % reductions, respectively. Sarker, et al. [77] found that the polyphenols contents in dark red bean protein hydrolysate were 44.3 mg GAE/g, and 38.89 mg GAE/g Roy, et al. [43]. The physicochemical properties of protein may affected by protein-phenolic interaction; the peptide activity may increase by blocking some amino acid side chains, thereby increasing the bioavailability and activity of polyphenols [78]. Polyphenols become entrapped in the peptide fragments and the enzyme hydrolyzes the protein, and becomes bound with polyphenols. Therefore, the polyphenols will be released and their free content will increase [79]. Additionally, it was found that alcalase hydrolysis of rice bran protein for 10 min was enough to extract all the bound phenolic acids [80].

#### 3.1.7. Antioxidant Activity

DPPH estimated the scavenging activity of protein hydrolysates and the results are presented in Figure 3A. The results herein showed that BKH (800 µg/mL) significantly scavenged a high level of DPPH˙ radical (95%). The concentration dependence of this phenomenon confirms the role of the phenolic compounds in this activity. The antioxidant activity of WKH, RKH, and BKH as determined by β-Carotene/linoleic acid bleaching, indicates that the inhibition in linoleic acid oxidation was increased with by the hydrolysate in a concentration-dependent manner. Figure 3B shows that BKH (800 µg/mL) has the highest values with 88% while RKH and WKH came in the second-order with 82 and 79% inhibition, respectively. The higher scavenging and antioxidant activity of this BKH hydrolysate may be attributed to the relatively high total polyphenols content in this protein hydrolysate. It was recently found that white kidney bean pepsin hydrolysate exhibited antioxidant activity with 85%, and white kidney bean papain hydrolysate exhibited 89%, respectively [21,75]. Furthermore, the dark red bean protein hydrolysate exhibited stronger antioxidant activity than the protein isolate and higher than ascorbic acid (AA), which is commercially used in the food industry [77]. The antioxidant mechanism of tested peptides depends on their content of aromatic amino acids donating an electron or transferring hydrogen to the free radicals for stability. Whereas, the acidic amino acids maintain the stability of the free radicals by giving a proton through the NH2 and COOH side chains [81]. Antioxidant peptides have critical importance in the food industry, where they keep the product quality by preventing the oxidation of proteins, lipids, and nucleic acids [82].

#### 3.1.8. Antimicrobial Activity of the Hydrolysates

##### Antibacterial

Table 4 and Figure 4 show the inhibition zone diameters (IZDs) of some pathogenic bacteria when subjected to BKH, RKH, and WKH at different concentrations (50, 100, 200, 400, and 800 µg/mL). BKH concentrations induced the largest IZDs in the range of 10–33 mm followed by RKH in the range of 12-31 mm. The G+ and G- bacteria; *S. pyogenes*, and *E. coli,* were the most sensitive to the protein hydrolysates, exhibiting 25 and 33 mm IZDs when subjected to 800 µg/mL. Alternatively, the G+ and G- bacteria, *L. monocytogenes*, *C. jejuni* were most resistant to the studied hydrolysates, recording only 26 and 21 mL IZDs, respectively. The lower IZDs observed in the resistant G- bacteria compared to the G+ ones might have resulted from the lipopolysaccharide layer in the membranes, which act as a block banning the antibacterial agents’ penetration. Besides, the presence of some enzymes in the periplasmic space of the G- bacteria may be capable of annealing foreign molecules [83].

The data show that BKH has the lowest MIC values (25–45 µg/mL) against the six studied bacteria, representing only less than 50% and 25% of the RKH, and WKH respective values, respectively. At higher concentrations, BKH could completely inhibit the bacteria recording MBC in the range 50–85 µg/mL against 110–175 µg/mL in the case of WKH. Roy, et al. [43] reported similar levels of IZDs of 20.26 mm and 19.23 mm, of red kidney beans hydrolysate against *Escherichia coli,* and *Pseudomonas aeruginosa,* respectively. Wahdan and Saad [75] found that white kidney bean protein hydrolysate produced by 4 h papain-hydrolysis inhibited the growth of tested bacteria with IZDs in the range of 70–90 µg/mL. The antibacterial action of peptides depends on bacterial species. First, the peptides may electrostatically bind to the bacterial membranes. Therefore, the intercellular structures and processes, such as cell wall synthesis, DNA, RNA, and protein synthesis could be affected and the plasma membrane may be interrupted [84].

##### Antifunga

The antifungal activity of white, red, and black Alcalase hydrolysates is presented in Table 5. The IZDs diameters of tested hydrolysates (50, 100, 200, 400, and 800 µg/mL) were in the range of (9–33 mm) against tested fungi, and in the range of (8–34 mm) against tested *Candida* and *Rhodotorula*. Therefore, there was no distinction of the action of the hydrolsates between fungi and *Candida*. The most sensitive fungi to the studied hydrolysates were *F. equiseti, F. avenaceum,* and *Candida, C. tropicalis* and *R. mucilginosa* were the most susceptible to the high concentration of kidney bean hydrolysate (800 µg/mL), showing IZDs in the range of 31–33, and 30–34 at 800 µg/mL, respectively. On the other hand, the fungi; *A. niger,* and *C. gleberta* seemed relatively the most resistant recoding IZDs around 23, and 25 mm against 800 µg/mL of BKH, respectively. The fungal growth was inhibited by BKH, marking a MIC range of 25–45 mm and eliminated at MFC in the range of 50–90 mm. The MIC and MFC of BKH and were higher than those of RKH and WKH (Table 6). Antimicrobial peptides (AMPs) have different action mechanisms, including the barrel-stave mechanism, which depends on the hydrophobicity of peptides that form pores in the microbial membrane by binding to the hydrophobic groups in the pore. The second is called the toroid pores or wormhole mechanisms or the carpet mechanism, where the polar peptides bind with phospholipids in membranes forming the toroid pores. Therefore, the peptides are deposited on the bilayer surface in a carpet-like fashion, mainly by ionic/electrostatic interactions, leading to cellular membrane destabilization and membrane destruction [85]. Since the protein hydrolysates logically encompass both hydrophobic and hydrophilic peptides, based on the variability of the constituting amino acids, both mechanisms may be functioning at the same time.

By comparison, of the values of MIC and MFC against the fungi with MIC and MBC against the studied bacteria (Table 6) refer to higher values in the case of the fungi. This difference may indicate that kidney protein hydrolysates are less effective against fungi than bacteria. The possible mechanisms indicate earlier suits more bacteria than fungi, whose biochemical structure is more sophisticated than the bacteria.

### 3.2. Chicken Meat Preservation with Kidney Bean Protein Hydrolysates

#### 3.2.1. Physicochemical Changes during Cold Storage

Table 7 presents the changes in some physicochemical parameters of raw chicken meat during cold storage for 30 days, i.e., pH, metmyoglobin (%), TVBN (mg N/100 g), and lipid oxidation inhibition (%), in response to graded additions of BKH, RKH and WKH. All parameters significantly increased with increasing the storage period from 0 to 10, 20, and 30 days, but lipid oxidation inhibition (%) significantly decreased. The pH values, metmyoglobin, and TVBN significantly decreased with the increments of BKH, RKH, and WKH concentrations (200, 400, and 800 µg/g) in a concentration-dependent manner. The pH value of the control sample increased from 5.6 to 8.9, i.e., reflecting a 40% relative increase after 30 days of cold storage. Supplementing meat with protein hydrolysate (800 µg/g) decreased pH by 25–30% of the control value after 30-day cold storage. The increase in the pH value with storage time (30 days) in the control samples coincided with parallel increases in total volatile basic nitrogen (TVBN), i.e., from 6.5 mg N/100 g at zero time to 8.35 mg N/100 g after 30 days of cold storage. This pH increase is probably a result of the metabolic activities of the contaminating spoilage bacteria, which can hydrolyze the proteins and lipids producing NH3 [86]. Treating the stored meat with the protein hydrolysate significantly reduced this value in a concentration-dependent manner, specially BKH (800 µg/g), which maintained the level of TVBN down to 6.50 mg N/100 g after 30 days of cold storage. This action is apparently due to the antibacterial activity of the protein hydrolysate previously evidenced. Similar results were observed by Saad, et al. [21], who reported that minced beef supplementation with pepsin kidney bean hydrolysate (100, and 200 µg/g) reduced the pH increase by 10%. The level of Met-myoglobin in the control sample increased rapidly with storage reaching 52.5% after 30 days of cold storage, i.e., exceeding the acceptable level in meat (40%). The addition of protein hydrolysates (800 µg/g) to chicken meat significantly slowed down the rate of meat deterioration, keeping the level of metmyoglobin down to 22–28 % after 30 days storage, i.e., achieving relative reductions of about 58–47 % of the control value, respectively. The highest reduction (58%) was achieved by BKH (800 µg/g). The increase in metmyoglobin with storage is evidently due to the auto-oxidation of meat protein and the oxidation of myoglobin to met-myoglobin, leading to undesirable meat color [87]. The reduction of this value by hydrolysate addition is probably due to their antioxidant action, previously quantified (Figure 3). Finally, the addition of protein hydrolysates to chicken meat at a high level (800 µg/g) significantly increased the efficiency of inhibiting lipid oxidation by about 59–70% over the control. The addition of protein hydrolysate to the preserved meat significantly inhibited the unwanted oxidative changes because of antioxidant activity, previously mentioned. Lipid oxidation is the main factor affecting the meat’s lifetime during storage. The formation of hydroperoxides and aldehydes are indicators of lipid oxidation [88]. Piñuel, et al. [89] found the addition of phaseolin isolated from red kidney bean to zebrafish inhibited the lipid oxidation by 82%, and Aslam, et al. [90] found that the addition of fish protein hydrolysate to chicken meat breast during storage delayed the lipid oxidation and all unwanted changes.

#### 3.2.2. Fluctuation in Color Parameters and Sensorial Traits of Raw Chicken Meat during 30 Days of Cold Storage

Figure 5 presents and predict the changes in the color parameters in chicken meat samples supplemented or un-supplemented with kidney bean protein hydrolysates after 30 days of cold storage. The whiteness indicator (*L**) of chicken meat significantly increased with WKH (800 µg/g) from 60.15 in control sample to 61.90, with a relative increase of 12% over control. However, RKH supplementation (800 µg/g) increased the redness (*a**) of chicken meat by 10% over the control, and BKH addition (800 µg/g) increased the blueness (*b**) of meat by 12% over the control. Meat whiteness is expected to increase with increasing WKH concentration, however decreased with increasing BKH concentration (Figure 5A–C). The results were confirmed by total color change (∆E) in Figure 6. The highest total color change showed in meat sample supplemented with WKH (400 µg/g), i.e., 2.4 with relative increase 25% about control. However the lowest value was found in samples supplemented with BKH (200 µg/g), i.e., 0.5. Generally, the color parameters faded after 30 days of storage, but supplementation with BKH, RKH and WKH significantly maintained about 75% of color attributes. Ab Aziz, et al. [91] found that all color parameters of chicken meat preserved at different temperature degrees decreased with the storage period. No available studies cast light on the changes in color parameters and sensorial traits of chicken meat samples supplemented with kidney bean protein hydrolysates. The addition of (BKH, RKH, and WKH) significantly maintained the meat quality, where BKH-supplemented chicken meat samples recorded the highest scores in tenderness and juiciness, i.e., 8.5, and 8.7 at zero-time storage (Table 8 and Figure 7), as it had the highest scores in water holding capacity in its original status. BKH also improved the taste and flavor of cooked chicken meat. A lower magnitude of meat quality improvement was recorded with RKH and WKH supplementation. Generally, all meat quality decreased after 30-day storage, but supplementation with the studied hydrolysates significantly reduced the deterioration rate by about 50–65% and reduction may be increased by increasing BKH concentration (Table 8 and Figure 7). It was observed that the sensory attributes of nuggets meat samples were significant during cold storage. However nuggets incorporated with GPP and GSCP had better shelf life and are highly acceptable than control [92].

#### 3.2.3. Microbial Changes in Chicken Meat during 30 Days Cold Storage

Table 9 and Figure 8 showed a significant *p* ≤ 0.05 increase in bacterial load during 30 days of cold storage. Supplementing chicken meat with a high concentration (800 µg/g) of BKH, RKH, and WKH significantly *p* ≤ 0.05 reduced the total bacterial count (mesophilic) and psychrophilic bacterial count by 35–45% over control after 30 days of cold storage. The potential antibacterial activity of the kidney bean protein hydrolysates could contribute to maintaining meat quality for about 20–30 days under refrigeration conditions. Saad, et al. [21] used pepsin kidney bean protein hydrolysate in maintaining minced beef quality by reducing microbial load by 22%. The acceptable total bacterial count in raw buffalo meat is less than <1 × 10^6^ CFU/g, based on Egyptian Standards (E.S.) No. 4334/2004 [93] following the International Commission on Microbiological Specification (ICMS, 1982). So, the BKH (800 µg/g)-enriched chicken may valid for human consumption after about 28–29 days of cold storage. While, chicken meat enriched with RKH (800 µg/g) is acceptable for consumption after 20 days of cold storage and WKH has the least shelf life, less than 20 days, and more than ten days of cold storage.

## 4. Conclusions

Legumes are valuable sources of bioactive peptides that can be used in various ways, including antimicrobial, antithrombotic, antioxidant and antihypertensive. Bioactive peptides can be easily and specifically generated by enzyme hydrolysis and microbial fermentation. The black, red, and white seed proteins can be transformed into valuable bioactive hydrolysate by a short time (30 min) Alcalase action. The hydrolysate products from the three protein sources also comprised adequate levels of total polyphenols, contributing to their global antioxidant and antimicrobial activities. Consequently, all prepared hydrolysates exhibited excellent antioxidant activities, particularly BKH showing the highest performance. The different hydrolysates (BKH, RKH, and WKH) manifested multiple functional properties like water and oil absorption capacity, emulsifying and foaming properties, especially around pH 4 and 5, which indicate their potential applications in low pH foods. BKH is a good candidate for meat preservation based on its capability to counteract lipid, protein and myoglobin oxidation and diminish bacterial and fungal contamination during cold storage. It can extend the cold shelf-life of the stored chicken meat up to 28–29 days while maintaining the sensorial properties to acceptable levels. Generally, the three products can be used in food systems as potential antioxidants and effective antibacterial in the following order of preference BKH > RKH > WKH.

## Figures and Tables

**Figure 1 molecules-26-04690-f001:**
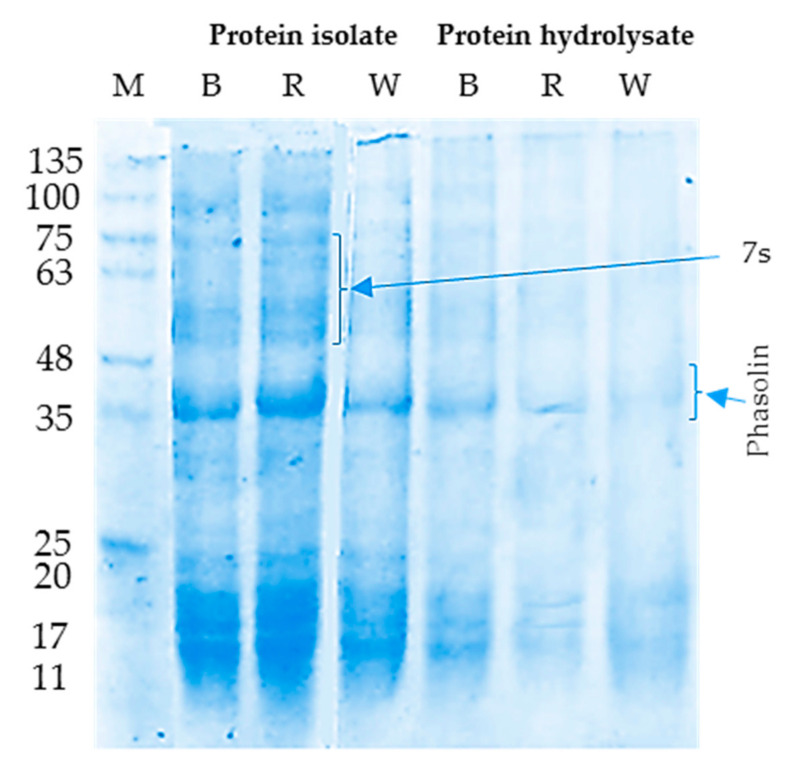
SDS-PAGE profile: (Lanes, 1–3) are for protein isolates and lanes 4–6 for protein hydrolysates (30 min Alcalase hydrolysis at 37 °C), isolated from black (B), red (R), and white (W) kidney bean seeds.M, molecular marker.

**Figure 2 molecules-26-04690-f002:**
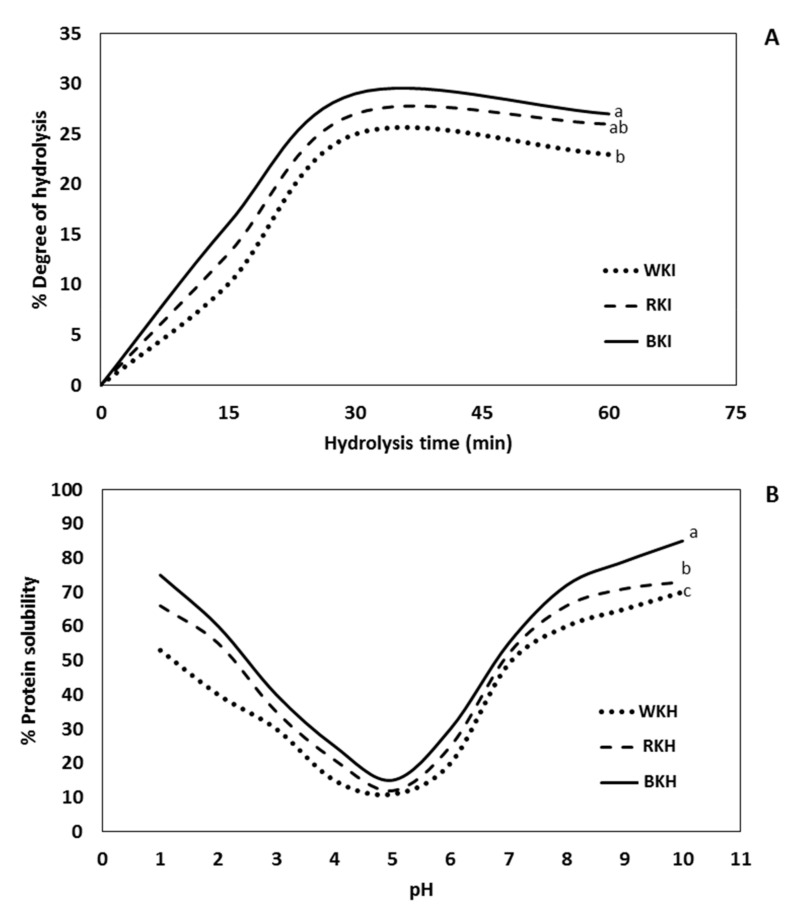
Hydrolysis degree (**A**) and pH-solubility curves (**B**) of black, red and white kidney seed protein hydrolysates (BKH, RKH and WKH) prepared by 30 min hydrolysis with Alcalase at 37 °C and pH 6. Means with different lowercase letters indicate significant differences at *p* ≤ 0.05 by LSD.

**Figure 3 molecules-26-04690-f003:**
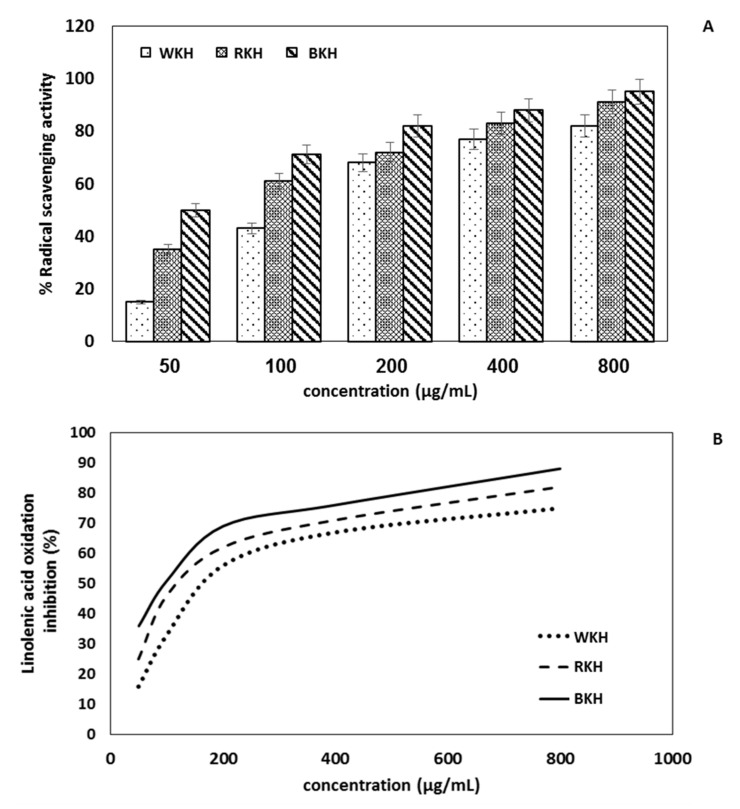
Free radical scavenging activity (**A**) and antioxidant activity (**B**) of protein Alcalase hydrolysates of black, red and white kidney bean protein (BKH, RKH, WKH). Protein Alcalase hydrolysis was conducted at 37 °C, pH 6 for 30 min. Means with different lowercase letters indicate significant differences at *p* ≤ 0.05 by LSD.

**Figure 4 molecules-26-04690-f004:**
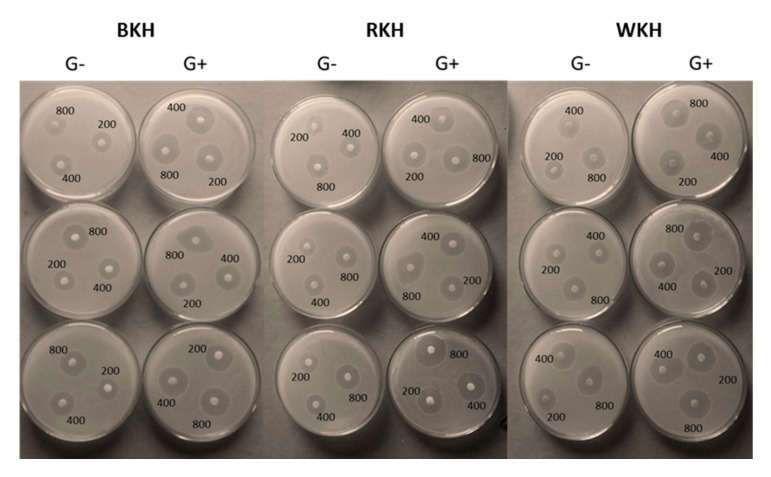
The antibacterial activity of black, red, and white kidney bean protein hydrolysate (BKH, RKH, and WKH) obtained by 30 min Alcalase hydrolysis against pathogenic G+, and G- bacteria.

**Figure 5 molecules-26-04690-f005:**
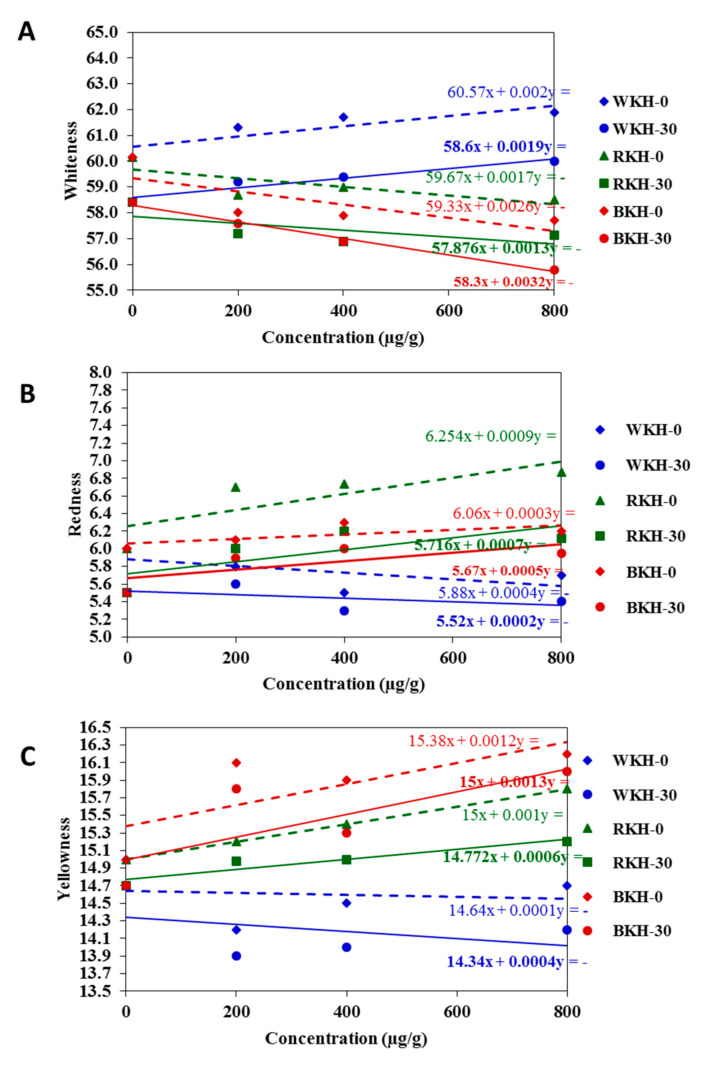
The changes in color parameters (**A**) Whiteness values, (**B**) Redness values, (**C**) yellowness values of supplemented meat with WKH, RKH, and BKH with increasing concentrations or storage period expressed by trendline regression curves for results prediction.

**Figure 6 molecules-26-04690-f006:**
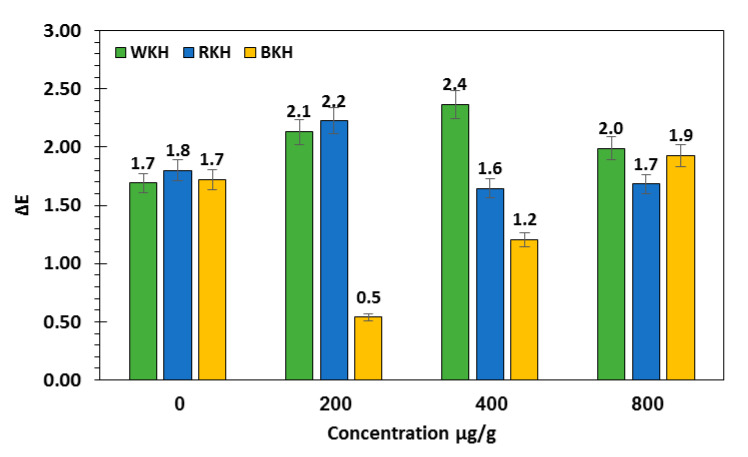
Total color change (∆E) of meat samples supplemented with kidney bean hydrolysates during storage period.

**Figure 7 molecules-26-04690-f007:**
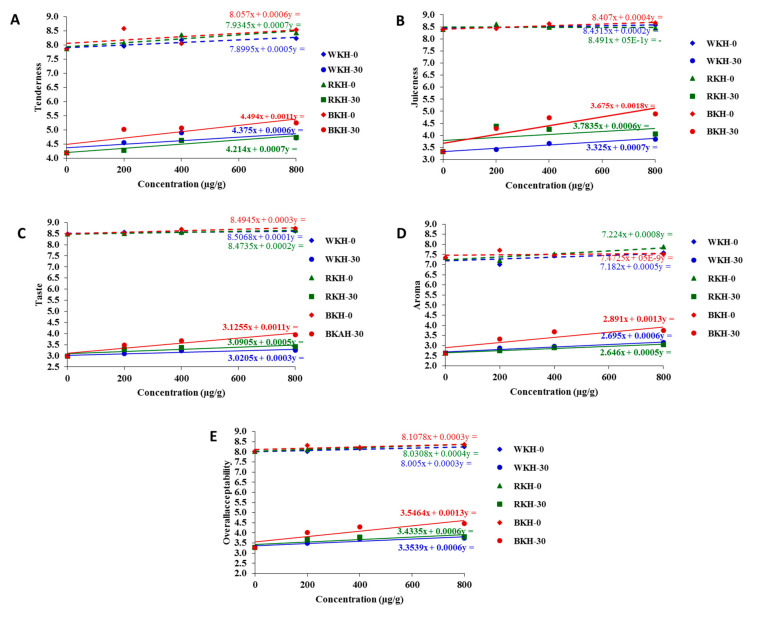
Expected changes in sensory properties; (**A**) Tenderness, (**B**) Juiciness, (**C**) Taste, (**D**) Aroma, (**E**) allover acceptability of cooked chicken meat supplemented with BKH, RKH, and WKH with increasing concentration or storage period expressed by trendline regression curves for results prediction.

**Figure 8 molecules-26-04690-f008:**
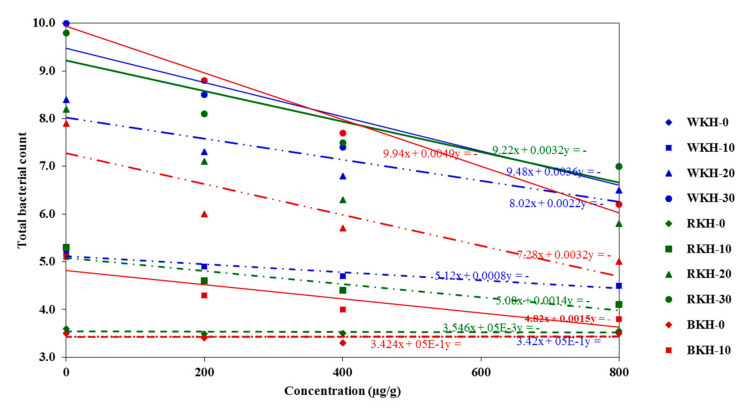
The bacterial load in chicken meat supplemented with BKH, RKH, and WKH during storage period expressed by trendline regression curves for results prediction.

**Table 1 molecules-26-04690-t001:** Chemical composition of total seed material (T), protein isolate (I) and protein hydrolysate (H), isolated from black, red, and white kidney bean seeds.

Chemical Composition (%)	Black Kidney Bean			Red Kidney Bean			White Kidney Bean		
Material Status	T	I	H	T	I	H	T	I	H
**Protein**	24.5 ± 0.1 ^c^	91 ± 1.1 ^a^	88 ± 0.7 ^ab^	22.7 ± 0.7 ^cd^	88.9 ± 1.2 ^ab^	85 ± 1 ^b^	21.3 ± 0.6 ^d^	85 ± 0.9 ^b^	82 ± 0.6 ^bc^
**Carbohydrates**	62.1 ± 0.3 ^ab^	32.5 ± 0.4 ^c^	4.7 ± 0.1 ^e^	65.1 ± 0.8 ^a^	36.7 ± 0.8 ^b^	8.7 ± 0.2 ^d^	64.2 ± 0.8 ^a^	37.2 ± 0.5 ^b^	8.3 ± 0.1 ^d^
**Fat**	2.1 ± 0.1 ^b^	ND	ND	2.5 ± 0.1 ^ab^	ND	ND	3.1 ± 0.1 ^a^	ND	ND
**Ash**	4.5 ± 0.3 ^d^	6.4 ± 0.1 ^b^	7.3 ± 0.2 ^ab^	4.19 ± 0.2 ^d^	5.7 ± 0.1 ^c^	6.5 ± 0.1 ^b^	5.1 ± 0.2 ^cd^	7.2 ± 0.2 ^ab^	9.4 ± 0.1 ^a^
**Moisture**	5.8 ± 0.1 ^b^	3.1 ± 0.1 ^cd^	ND	6.14 ± 0.2 ^ab^	3.5 ± 0.1 ^c^	0.88 ± 0.03 ^e^	6.45 ± 0.1 ^a^	1.5 ± 0.01 ^d^	0.98 ± 0.02 ^e^

ND, not detected. All values are expressed as means ± SD; Means with different lowercase letters in the same row indicate significant differences at *p* ≤ 0.05 by LSD. T is the total material. I is the protein isolate. H (protein hydrolysate) is obtained after 30-min hydrolysis with Alcalase at 37 °C.

**Table 2 molecules-26-04690-t002:** Functional properties of black, red and white kidney bean protein hydrolysate (BKH, RKH, and WKH) obtained after 30 min Alcalase hydrolysis at 37 °C.

Protein Hydrolysate		Functional Properties				
	HT (min)	WAC (g/g)	OAC (g/g)	FS (min)		
				pH 3	pH 5	pH 7
BKH	0	6.9 ± 0.5 ^d^	6.8 ± 0.2 ^d^	23 ± 0.1 ^c^	65 ± 0.3 ^c^	26 ± 0.6 ^c^
	30	9.4 ± 0.2 ^a^	10.5 ± 0.6 ^a^	27 ± 0.3 ^a^	88 ± 0.9 ^a^	30 ± 0.8 ^a^
RKH	0	6.7 ± 0.4 ^de^	6.5 ± 0.3 ^de^	20 ± 0.8 ^d^	44 ± 0.8 ^e^	23 ± 0.2 ^d^
	30	8.2 ± 0.8 ^b^	9.3 ± 0.1 ^b^	25 ± 0.6 ^b^	75 ± 0.7 ^b^	28 ± 0.5 ^b^
WKH	0	6.1 ± 0.3 ^e^	6.0 ± 0.1 ^e^	18 ± 0.9 ^e^	29 ± 0.1 ^f^	21 ± 0.4 ^e^
	30	7.2 ± 0.1 ^c^	8.1 ± 0.2 ^c^	22 ± 0.7 ^cd^	50 ± 0.9 ^d^	25 ± 0.3 ^cd^

HT, hydrolysis time; WAC, water-absorbing capacity; OAC, oil-absorbing capacity; FS, foaming stability. All values are expresses as means ± SD. Means with different lowercase letters in the same column indicate significant differences at *p* ≤ 0.05 by LSD.

**Table 3 molecules-26-04690-t003:** Total phenolic, total flavonoids and total anthocyanin content in different concentrations of black, red and white kidney bean protein hydrolysates (BKH, RKH, and WKH) obtained after 30 min Alcalase hydrolysis at 37 °C.

Concn (µg/mL)	Total Phenolic (mg GAE/g)			Total Flavonoids (mg QE/g)			Total Anthocyanin (µg/mL)		
	BKH	RKH	WKH	BKH	RKH	WKH	BKH	RKH	WKH
**50**	11.2 ± 0.1 ^e^	9.5 ± 0.5 ^e^	7.2 ± 0.9 ^e^	6.5 ± 0.1 ^e^	5.2 ± 0.1 ^e^	6.2 ± 0.3 ^d^	1.90 ± 0.1 ^d^	1.02 ± 0.01 ^e^	0.11 ± 0.02 ^d^
**100**	17.4 ± 0.5 ^d^	15.4 ± 0.4 ^d^	12.4 ± 0.8 ^d^	8.7 ± 0.2 ^d^	7.1 ± 0.3 ^d^	8.1 ± 0.2 ^c^	2.56 ± 0.2 ^c^	1.56 ± 0.09 ^d^	0.25 ± 0.01 ^c^
**200**	31.6 ± 0.7 ^c^	28.7 ± 0.3 ^c^	24.3 ± 0.6 ^c^	11.9 ± 0.1 ^c^	10 ± 0.2 ^c^	12.2 ± 0.4 ^b^	4.5 ± 0.2 ^bc^	2.45 ± 0.1 ^c^	0.30 ± 0.01 ^bc^
**400**	39.6 ± 1.1 ^b^	35.5 ± 0.2 ^b^	33.8 ± 0.7 ^b^	14.1 ± 0.4 ^b^	12.5 ± 0.5 ^b^	13.2 ± 0.7 ^ab^	4.77 ± 0.3 ^b^	2.89 ± 0.1 ^b^	0.34 ± 0.03 ^b^
**800**	63.3 ± 1.5 ^a^	60.6 ± 0.9 ^a^	55.8 ± 1.2 ^a^	16.2 ± 0.8 ^a^	15.8 ± 0.3 ^a^	14.4 ± 0.6 ^a^	5.6 ± 0.4 ^a^	3.12 ± 0.2 ^a^	0.8 ± 0.04 ^a^

All values are expressed as means ± SD. Means with different lowercase letters in the same column indicate significant differences at *p* ≤ 0.05 by LSD.

**Table 4 molecules-26-04690-t004:** Antibacterial activity against gram-positive and gram-negative bacterial strains (expressed as the diameter of the inhibition zones) of different concentrations (50–800 µg/mL) of black, red and white kidney bean protein hydrolysate (BKH, RKH and WKH) obtained after 30 min Alcalase hydrolysis at 37 °C.

Bacteria	BKH					RKH					WKH				
	50	100	200	400	800	50	100	200	400	800	50	100	200	400	800
**G+**															
*B. cereus*	15 ± 0.0 ^ab^	21 ± 0.14 ^b^	23 ± 0.21 ^b^	26 ± 0.16 ^b^	28 ± 0.11 ^b^	14 ± 0.11 ^ab^	19 ± 0.15 ^b^	22 ± 0.14 ^b^	24 ± 0.12 ^b^	27 ± 0.21 ^b^	-	15 ± 0.25 ^b^	21 ± 0.21 ^ab^	23 ± 0.12 ^b^	27 ± 0.33 ^b^
*L. monocytogenes*	14 ± 0.11 ^b^	19 ± 0.22 ^c^	21 ± 0.14 ^c^	25 ± 0.14 ^bc^	27 ± 0.17 ^bc^	13 ± 0.34 ^b^	18 ± 0.19 ^bc^	21 ± 0.15 ^bc^	23 ± 0.18 ^bc^	26 ± 0.28 ^bc^	-	13 ± 0.14 ^c^	19 ± 0.15 ^b^	22 ± 0.14 ^bc^	26 ± 0.14 ^bc^
*S. pyogenes*	16 ± 0.14 ^a^	23 ± 0.21 ^a^	26 ± 0.19 ^a^	29 ± 0.15 ^a^	33 ± 0.0 ^a^	15 ± 0.22 ^a^	22 ± 0.23 ^a^	24 ± 0.19 ^a^	27 ± 0.23 ^a^	31 ± 0.19 ^a^	-	17 ± 0.16 ^a^	22 ± 0.17 ^a^	25 ± 0.18 ^a^	30 ± 0.25 ^a^
**G-**															
*E. coli*	13 ± 0.18 ^bc^	18 ± 0.15 ^cd^	20 ± 0.0 ^cd^	23 ± 0.22 ^c^	25 ± 0.23 ^c^	-	16 ± 0.14 ^c^	19 ± 0.22 ^c^	21 ± 0.17 ^c^	24 ± 0.17 ^c^	-	12 ± 0.18 ^cd^	18 ± 0.19 ^bc^	21 ± 0.19 ^c^	25 ± 0.41 ^c^
*C. jejuni*	10 ± 0.21 ^d^	14 ± 0.12 ^e^	16 ± 0.21 ^e^	20 ± 0.23 ^e^	22 ± 0.12 ^e^	-	12 ± 0.51 ^e^	15 ± 0.31 ^e^	18 ± 0.55 ^e^	20 ± 0.15 ^e^	-	10 ± 0.32 ^e^	13 ± 0.23 ^d^	18 ± 0.21 ^d^	21 ± 0.23 ^e^
*S. typhi*	11 ± 0.13 ^c^	16 ± 0.21 ^d^	19 ± 0.32 ^d^	21 ± 0.22 ^d^	23 ± 0.17 ^d^	-	14 ± 0.14 ^d^	17 ± 0.32 ^d^	19 ± 0.21 ^d^	21 ± 0.14 ^d^	-	11 ± 0.47 ^d^	15 ± 0.45 ^c^	20 ± 0.33 ^cd^	23 ± 0.12 ^d^

All values are expresses as means ± SD. Means with different lowercase letters in the same column indicate significant differences at *p* ≤ 0.05 by LSD.

**Table 5 molecules-26-04690-t005:** Antifungal activity of black kidney, red and white bean protein hydrolysate (BKH, RKH, and WKH) obtained after 30 min Alcalase hydrolysis at 37 °C, against yeast and fungal strains (means ± SD).

Microorganisms	BKH					RKH					WKH				
	50	100	200	400	800	50	100	200	400	800	50	100	200	400	800
**Fungi**															
***A. niger***	9 ± 0.0 ^d^	13 ± 0.2 ^d^	14 ± 0.2 ^d^	18 ± 0.1 ^e^	23 ± 0.2 ^e^	-	10 ± 0.1 ^d^	12 ± 0.3 ^d^	16 ± 0.2 ^d^	21 ± 0.2 ^f^	-	-	10 ± 0.1 ^d^	13 ± 0.2 ^d^	18 ± 0.2 ^f^
***A. fumigatus***	11 ± 0.1 ^c^	15 ± 0.4 ^c^	16 ± 0.1 ^bc^	21 ± 0.3 ^d^	27 ± 0.1 ^d^	-	12 ± 0.2 ^c^	14 ± 1 ^c^	19 ± 0.3 ^c^	25 ± 0.4 ^d^	-	-	12 ± 0.2 ^bc^	16 ± 0.1 ^c^	22 ± 0.1 ^d^
***A. flavus***	10 ± 0.2 ^c^	14 ± 0.3 ^c^	15 ± 0.4 ^c^	20 ± 0.1 ^de^	24 ± 0.8 ^e^	-	13 ± 0.6 ^bc^	14 ± 0.4 ^c^	18 ± 0.8 ^cd^	22 ± 0.7 ^e^	-	-	11 ± 0.4 ^c^	15 ± 0.5 ^cd^	19 ± 0.4 ^e^
***P. chrysogenum***	12 ± 0.4 ^b^	16 ± 0.5 ^b^	17 ± 0.8 ^b^	23 ± 0.4 ^c^	29 ± 0.7 ^c^	-	14 ± 0.4 ^b^	15 ± 0.5 ^b^	21 ± 0.7 ^bc^	27 ± 0.3 ^c^	-	-	13 ± 0.5 ^b^	18 ± 0.1 ^bc^	24 ± 0.5 ^c^
***F. equiseti***	15 ± 0.7 ^a^	19 ± 0.7 ^a^	20 ± 0.7 ^a^	26 ± 0.5 ^a^	33 ± 0.9 ^a^	-	17 ± 0.3 ^a^	18 ± 0.7 ^a^	24 ± 0.9 ^a^	31 ± 0.5 ^a^	-	-	16 ± 0.7 ^a^	21 ± 0.2 ^a^	28 ± 0.8 ^a^
***F. avenaceum***	14 ± 0.6 ^ab^	18 ± 0.1 ^ab^	19 ± 0.3 ^ab^	24 ± 0.3 ^b^	31 ± 0.5 ^b^	-	16 ± 0.1 ^ab^	17 ± 0.6 ^ab^	22 ± 0.1 ^b^	29 ± 0.4 ^b^	-	-	15 ± 0.1 ^ab^	19 ± 0.4 ^b^	26 ± 0.4 ^b^
**Candida**															
***C. gelbeta***	9 ± 0.1 ^d^	12 ± 0.4 ^d^	15 ± 0.2 ^e^	18 ± 0.5 ^e^	25 ± 0.1 ^e^	-	10 ± 0.5 ^c^	12 ± 0.1 ^e^	16 ± 0.5 ^d^	22 ± 0.4 ^e^	-	8 ± 0.1 ^d^	10 ± 0.1 ^e^	14 ± 0.8 ^d^	18 ± 0.1 ^e^
***C. tropicalis***	11 ± 0.3 ^c^	15 ± 0.2 ^bc^	19 ± 0.0 ^c^	22 ± 0.1 ^cd^	30 ± 0.2 ^c^	-	12 ± 0.1 ^bc^	15 ± 0.2 ^c^	20 ± 0.1 ^bc^	27 ± 0.7 ^c^	-	10 ± 0.4 ^bc^	13 ± 0.4 ^cd^	18 ± 0.7 ^bc^	23 ± 0.5 ^c^
***C. albicans***	10 ± 0.1 ^cd^	13 ± 0.9 ^c^	17 ± 0.3 ^d^	21 ± 0.2 ^d^	28 ± 0.8 ^d^	-	11 ± 0.2 ^c^	14 ± 0.7 ^d^	19 ± 0.2 ^c^	25 ± 0.4 ^d^	-	9 ± 0.1 ^c^	12 ± 0.5 ^d^	17 ± 0.6 ^c^	21 ± 0.3 ^d^
***R. rubra***	14 ± 0.6 ^ab^	17 ± 0.7 ^ab^	21 ± 0.4 ^b^	25 ± 0.4 ^b^	32 ± 0.7 ^b^	13 ± 0.1 ^ab^	15 ± 0.7 ^ab^	18 ± 0.9 ^b^	22 ± 0.4 ^ab^	29 ± 0.2 ^b^	-	13 ± 0.5 ^ab^	16 ± 0.2 ^b^	20 ± 0.1 ^ab^	25 ± 0.6 ^b^
***R. minuta***	13 ± 0.4 ^b^	16 ± 0.6 ^b^	20 ± 0.1 ^bc^	23 ± 0.9 ^c^	31 ± 0.1 ^bc^	12 ± 0.5 ^b^	13 ± 0.6 ^b^	16 ± 0.6 ^c^	21 ± 0.4 ^b^	28 ± 0.5 ^bc^	-	11 ± 0.1 ^b^	14 ± 0.7 ^c^	19 ± 0.2 ^b^	24 ± 0.4 ^bc^
***R.mucilginosa***	15 ± 0.9 ^a^	18 ± 0.5 ^a^	23 ± 0.9 ^a^	27 ± 0.0 ^a^	34 ± 0.2 ^a^	14 ± 0.4 ^a^	16 ± 0.9 ^a^	20 ± 0.4 ^a^	23 ± 0.1 ^a^	31 ± 0.1 ^a^	-	14 ± 0.4 ^a^	18 ± 0.2 ^a^	21 ± 0.7 ^a^	27 ± 0.7 ^a^

ND, not detected, All values are expressed as the means ± SD. Means with different lowercase letters in the same column indicate significant differences at *p* ≤ 0.05 by LSD.

**Table 6 molecules-26-04690-t006:** Minimum inhibitory concentration (MIC), minimum bactericidal concentration (MBC), and minimum fungicidal concentration (MFC) of black, red and white kidney bean protein hydrolysate (BKH, RKH and WKH) obtained after 30 min Alcalase hydrolysis at 37 °C, against different microbial strains.

Microorganisms	BKH		RKH		WKH	
***Bacteria***	**MIC**	**MBC**	**MIC**	**MBC**	**MIC**	**MBC**
***Bacillus cereus***	30 ^e^	55 ^e^	40 ^e^	80 ^e^	65 ^e^	120 ^e^
***listeria monocytogenes***	35 ^d^	65 ^d^	45 ^d^	85 ^d^	70 ^d^	130 ^d^
***Streptococcus pyogenes***	25 ^f^	50 ^f^	35 ^f^	65 ^f^	60 ^f^	110 ^f^
***Escherichia coli***	40 ^c^	75 ^c^	65 ^c^	125 ^c^	75 ^c^	140 ^c^
***Campylobacter jejuni***	45 ^a^	85 ^a^	75 ^a^	145 ^a^	90 ^a^	175 ^a^
***Salmonella typhi***	40 ^b^	75 ^b^	70 ^b^	130 ^b^	80 ^b^	155 ^b^
**Fungi**	**MIC**	**MFC**	**MIC**	**MFC**	**MIC**	**MFC**
***Aspergillus niger***	45 ^a^	90 ^a^	90 ^a^	170 ^a^	180 ^a^	320 ^a^
***Aspergillus fumigatus***	35 ^c^	70 ^c^	75 ^c^	140 ^c^	140 ^c^	270 ^c^
***Aspergillus flavus***	40 ^b^	75 ^b^	80 ^b^	150 ^b^	150 ^b^	280 ^b^
***Penicillium chrysogenum***	30 ^d^	55 ^d^	70 ^d^	130 ^d^	135 ^d^	250 ^d^
***Fusarium equiseti***	25 ^f^	50 ^f^	60 ^f^	110 ^f^	110 ^f^	200 ^f^
***Fusarium avenaceum***	30 ^e^	55 ^e^	65 ^e^	120 ^e^	120 ^e^	220 ^e^
**Candida**						
***Candia gelbeta***	45 ^a^	90 ^a^	80 ^a^	150 ^a^	90 ^a^	170 ^a^
***Candida tropicalis***	35 ^c^	70 ^c^	65 ^c^	120 ^c^	80 ^c^	150 ^c^
***Candida albicans***	40 ^b^	75 ^b^	70 ^b^	130 ^b^	85 ^b^	160 ^b^
***Rhodotorula rubra***	25 ^e^	50 ^e^	40 ^e^	70 ^e^	65 e	120 ^e^
***Rhodotorula minuta***	30 ^d^	55 ^d^	45 ^d^	80 ^d^	70 ^d^	130 ^d^
***Rhodotorula mucilginosa***	**20** ^f^	**40** ^f^	30 ^f^	60 ^f^	55 ^f^	100 ^f^

Means with different lowercase letters in the same column indicate significant differences at *p* ≤ 0.05 by LSD.

**Table 7 molecules-26-04690-t007:** Physicochemical changes in raw chicken meat supplemented with black, red, and white kidney bean protein hydrolysate (BKH, RKH, and WKH) obtained by alcalase after 30 min hydrolysis, at 0–30 days cold storage at 4 °C.

Sample	Concn (µg/g)	Duration of Cold Storage (Days)															
		0	10	20	30	0	10	20	30	0	10	20	30	0	10	20	30
		pH				TVBN (mg N/100 g Sample)				Metmyoglobin (%)				Lipid Oxidation Inhibition (%)			
Cont	0.0	5.60 ^a^	6.57 ^a^	7.50 ^a^	8.90 ^a^	6.50 ^a^	7.20 ^a^	7.80 ^a^	8.35 ^a^	9.00 ^a^	28.00 ^a^	42.00 ^a^	52.50 ^a^	33 ^ab^	24 ^c^	18 ^c^	9 ^d^
BKH	200	5.45 ^b^	5.90 ^c^	6.45 ^c^	7.00 ^c^	6.22 ^c^	6.50 ^c^	6.95 ^c^	7.20 ^c^	8.99 ^a^	18.00 ^c^	21.00 ^b^	30.00 ^b^	33 ^ab^	29 ^b^	27 ^c^	17 ^c^
	400	5.31 ^b^	5.70 ^c^	6.20 ^c^	6.82 ^d^	6.10 ^d^	6.20 ^c^	6.45 ^c^	6.80 ^d^	8.15 ^b^	13.90 ^d^	16.00 ^c^	25.00 ^c^	33 ^ab^	31 ^b^	29 ^b^	25 ^b^
	800	5.19 ^c^	5.66 ^c^	5.90 ^d^	6.24 ^d^	5.56 ^e^	5.86 ^d^	6.30 ^c^	6.50 ^d^	7.51 ^c^	11.30 ^d^	14.55 ^c^	22.60 ^c^	34 ^a^	33 ^a^	32 ^a^	30 ^a^
RKH	200	5.48 ^b^	6.00 ^b^	7.00 ^b^	7.41 ^b^	6.34 ^b^	6.80 ^b^	7.20 ^b^	7.40 ^c^	8.8 ^a^	20.30 ^b^	25.90 ^b^	35.00 ^b^	33 ^ab^	27 ^c^	25 ^c^	14 ^c^
	400	5.34 ^b^	5.70 ^c^	6.50 ^c^	6.90 ^d^	6.19 ^c^	6.40 ^c^	6.90 ^c^	7.15 ^c^	8.3 ^b^	16.10 ^c^	23.10 ^b^	31.50 ^b^	34 ^a^	30 ^b^	28 ^b^	21 ^c^
	800	5.29 ^c^	5.50 ^c^	6.20 ^c^	6.50 ^d^	6.11 ^d^	6.31 ^c^	6.50 ^c^	6.80 ^d^	7.9 ^c^	13.30 ^d^	16.80 ^c^	27.30 ^c^	35 ^a^	32 ^ab^	30 ^b^	28 ^b^
WKH	200	5.56 ^ab^	6.20 ^b^	7.10 ^b^	7.65 ^b^	6.40 ^ab^	7.00 ^ab^	7.40 ^b^	7.90 ^b^	8.90 ^a^	22.40 ^b^	28.00 ^b^	37.80 ^b^	35 ^a^	25 ^c^	23 ^c^	11 ^d^
	400	5.45 ^b^	6.00 ^b^	6.80 ^c^	7.10 ^c^	6.35 ^b^	6.70 ^b^	7.00 ^b^	7.20 ^c^	8.5 ^ab^	17.50 ^c^	24.50 ^b^	31.50 ^b^	34 ^a^	27 ^c^	25 ^c^	19 ^c^
	800	5.33 ^b^	5.80 ^c^	6.20 ^c^	6.78 ^d^	6.21 ^c^	6.45 ^b^	6.80 ^c^	7.00 ^c^	8.2 ^b^	14.70 ^d^	20.30 ^c^	28.00 ^c^	35 ^a^	29 ^b^	28 ^b^	22 ^c^

TVBN: Total volatile basic nitrogen. Means with different lowercase letters in the same column indicate significant differences at *p* ≤ 0.05 by LSD.

**Table 8 molecules-26-04690-t008:** Sensory properties of cooked chicken meat supplemented with black, red, and white kidney bean protein hydrolysate (BKH, RKH, and WKH) obtained by 30 min Alcalase hydrolysis, at 0 and 30 days cold storage at 4 °C.

Sample	Concn (µg/g)	Storage (Day)									
		0	30	0	30	0	30	0	30	0	30
		Tenderness		Juiciness		Taste		Aroma		Overall Acceptability	
**Cont.**	0	7.9 ± 0.1 ^a^	4.2 ± 0.2 ^b^	8.4 ± 0.2 ^a^	3.3 ± 0.2 ^b^	8.5 ± 0.1 ^a^	3.0 ± 0.2 ^b^	7.4 ± 0.6 ^a^	2.6 ± 0.5 ^b^	8.0 ± 0.1 ^a^	3.3 ± 0.2 ^b^
**BKH**	200	8.6 ± 0.3 ^a^	5.0 ± 0.3 ^b^	8.4 ± 0.4 ^a^	4.3 ± 0.6 ^b^	8.5 ± 0.1 ^a^	3.5 ± 0.4 ^b^	7.7 ± 0.7 ^a^	3.3 ± 0.3 ^b^	8.3 ± 0.2 ^a^	4.0 ± 0.7 ^b^
	400	8.1 ± 0.2 ^a^	5.1 ± 0.4 ^b^	8.6 ± 0.3 ^a^	4.7 ± 0.7 ^b^	8.7 ± 0.3 ^a^	3.7 ± 0.8 ^b^	7.4 ± 0.2 ^a^	3.7 ± 0.6 ^b^	8.2 ± 0.1 ^a^	4.3 ± 0.1 ^b^
	800	8.5 ± 0.4 ^a^	5.3 ± 0.5 ^b^	8.7 ± 0.3 ^a^	4.9 ± 0.8 ^b^	8.7 ± 0.3 ^a^	3.9 ± 0.6 ^b^	7.5 ± 0.5 ^a^	3.7 ± 0.5 ^b^	8.4 ± 0.6 ^a^	4.5 ± 0.3 ^b^
**RKH**	200	8.1 ± 0.5 ^a^	4.3 ± 0.3 ^b^	8.6 ± 0.2 ^a^	4.4 ± 0.4 ^b^	8.5 ± 0.2 ^a^	3.3 ± 0.3 ^b^	7.2 ± 0.3 ^a^	2.7 ± 0.5 ^b^	8.1 ± 0.6 ^a^	3.7 ± 0.5 ^b^
	400	8.3 ± 0.2 ^a^	4.6 ± 0.1 ^b^	8.5 ± 0.2 ^a^	4.3 ± 0.3 ^b^	8.6 ± 0.3 ^a^	3.4 ± 0.1 ^b^	7.5 ± 0.4 ^a^	2.9 ± 0.4 ^b^	8.2 ± 0.7 ^a^	3.8 ± 0.9 ^b^
	800	8.4 ± 0.2 ^a^	4.7 ± 0.2 ^b^	8.5 ± 0.3 ^a^	4.1 ± 0.5 ^b^	8.7 ± 0.2 ^a^	3.4 ± 0.3 ^b^	7.9 ± 0.6 ^a^	3.0 ± 0.2 ^b^	8.4 ± 0.2 ^a^	3.8 ± 0.3 ^b^
**WKH**	200	8.0 ± 0.2 ^a^	4.6 ± 0.3 ^b^	8.5 ± 0.3 ^a^	3.4 ± 0.5 ^b^	8.6 ± 0.2 ^a^	3.1 ± 0.1 ^b^	7.0 ± 0.5 ^a^	2.9 ± 0.2 ^b^	8.0 ± 0.1 ^a^	3.5 ± 0.6 ^b^
	400	8.2 ± 0.3 ^a^	4.9 ± 0.1 ^b^	8.5 ± 0.2 ^a^	3.7 ± 0.4 ^b^	8.5 ± 0.1 ^a^	3.2 ± 0.2 ^b^	7.4 ± 0.7 ^a^	3.0 ± 0.4 ^b^	8.2 ± 0.2 ^a^	3.7 ± 0.4 ^b^
	800	8.2 ± 0.1 ^a^	4.7 ± 0.2 ^b^	8.6 ± 0.1 ^a^	3.9 ± 0.2 ^b^	8.6 ± 0.1 ^a^	3.3 ± 0.1 ^b^	7.6 ± 0.6 ^a^	3.2 ± 0.3 ^b^	8.2 ± 0.4 ^a^	3.7 ± 0.8 ^b^

Cont.: control. All values are expressed as the means ± SD. Means with different lowercase letters in the same column indicate significant differences at *p* ≤ 0.05 by LSD.

**Table 9 molecules-26-04690-t009:** Total bacterial count (TBC) and psychrophilic bacterial count (PBC) of raw chicken meat, at 0 and 30 days cold storage at 4 °C, as supplemented with black, red, and white kidney bean protein hydrolysate (BKH, RKH, and WKH) obtained by 30 min Alcalase hydrolysis.

Sample	Concn (µg/g)	Storage (Day)							
		0	10	20	30	0	10	20	30
		Total Bacterial Count (Log CFU/g)				Psychrophilic Bacterial Count (Log CFU/g)			
**BKH**	**0**	3.50 ± 0.17 ^a^	5.10 ± 0.19 ^a^	7.9 ± 0.22 ^a^	10.2 ± 0.14 ^a^	2.80 ± 0.17 ^a^	4.50 ± 0.23 ^ab^	7.50 ± 0.18 ^a^	9.20 ± 0.12 ^a^
	**200**	3.31 ± 0.15 ^b^	4.30 ± 0.21 ^b^	6.0 ± 0.24 ^c^	8.8 ± 0.17 ^b^	2.66 ± 0.19 ^b^	3.90 ± 0.25 ^c^	6.20 ± 0.19 ^b^	8.50 ± 0.19 ^ab^
	**400**	3.25 ± 0.15 ^b^	4.00 ± 0.12 ^b^	5.7 ± 0.21 ^d^	7.7 ± 0.14 ^c^	2.41 ± 0.22 ^c^	3.50 ± 0.31 ^c^	5.90 ± 0.21 ^c^	7.40 ± 0.15 ^b^
	**800**	3.18 ± 0.11 ^b^	3.80 ± 0.15 ^c^	5.0 ± 0.28 ^d^	6.2 ± 0.22 ^d^	2.33 ± 0.31 ^c^	3.10 ± 0.33 ^c^	4.90 ± 0.39 ^d^	6.00 ± 0.17 ^c^
**RKH**	**0**	3.60 ± 0.21 ^a^	5.30 ± 0.18 ^a^	8.2 ± 0.31 ^a^	9.8 ± 0.21 ^a^	2.75 ± 0.18 ^a^	4.30 ± 0.25 ^b^	6.90 ± 0.22 ^ab^	8.40 ± 0.32 ^a^
	**200**	3.33 ± 0.14 ^b^	4.60 ± 0.13 ^b^	7.1 ± 0.25 ^b^	8.1 ± 0.23 ^b^	2.61 ± 0.21 ^b^	4.00 ± 0.27 ^b^	6.10 ± 0.31 ^b^	7.90 ± 0.35 ^b^
	**400**	3.29 ± 0.15 ^b^	4.40 ± 0.14 ^b^	6.3 ± 0.21 ^c^	7.5 ± 0.33 ^c^	2.55 ± 0.16 ^b^	3.80 ± 0.31 ^c^	5.80 ± 0.18 ^c^	7.10 ± 0.18 ^b^
	**800**	3.21 ± 0.13 ^b^	4.10 ± 0.18 ^b^	5.8 ± 0.28 ^d^	7.0 ± 0.11 ^c^	2.45 ± 0.18 ^c^	3.50 ± 0.21 ^c^	5.30 ± 0.23 ^c^	6.50 ± 0.22 ^c^
**WKH**	**0**	3.50 ± 0.15 ^a^	5.20 ± 0.14 ^a^	8.4 ± 0.21 ^a^	10.0 ± 0.11 ^a^	2.80 ± 0.12 ^a^	4.70 ± 0.22 ^a^	7.10 ± 0.12 ^a^	8.90 ± 0.21 ^ab^
	**200**	3.40 ± 0.18 ^b^	4.90 ± 0.12 ^ab^	7.3 ± 0.25 ^b^	8.5 ± 0.12 ^b^	2.79 ± 0.15 ^a^	4.40 ± 0.24 ^ab^	6.80 ± 0.15 ^ab^	7.40 ± 0.32 ^b^
	**400**	3.30 ± 0.17 ^b^	4.70 ± 0.15 ^b^	6.8 ± 0.27 ^c^	7.4 ± 0.15 ^c^	2.62 ± 0.14 ^b^	4.20 ± 0.27 ^b^	6.10 ± 0.19 ^b^	7.00 ± 0.28 ^b^
	**800**	3.20 ± 0.16 ^b^	4.50 ± 0.14 ^b^	6.5 ± 0.23 ^c^	7.0 ± 0.19 ^c^	2.59 ± 0.16 ^b^	4.00 ± 0.23 ^b^	5.80 ± 0.21 ^c^	6.70 ± 0.22 ^c^

All values are expressed as the means ± SD. Means with different lowercase letters in the same column indicate significant differences at *p* ≤ 0.05 by LSD.

## Data Availability

Not applicable.

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
