# Peer review of "Biochemical and Functional Characterization of Kidney Bean Protein Alcalase-Hydrolysates and Their Preservative Action on Stored Chicken Meat"

_molecules, 2021, doi:10.3390/molecules26154690_

Round 1

Reviewer 1 Report

This manuscript describes in detail a study of using protein hydrolysates as antimicrobails for meat shelf-life extension. In my opinion, the manuscript can be published with minor changes such as the following:

  • The title should be changed and include the biggest part of the study that is relevant to the characterization of the hydrolysates. Rephrase it so as to be easier to track the article and find information when searching for info relevant to the hydrolysates you studied.
  • Table 5 needs to be improved. 
  • The results of Table 8, in my opinion does not significantly help the reader. Only 0 and 30 days results are presented and since the meat is studied and has a big variability in colour, I think that the authors should present the Total colour change (ΔΕ) index vs storage time. It would be easier and clearer for the reader to check whether there is a positive effect of using the hydrolysates and which one and at what concentration.
  • In general, the authors did not model any of their results. If the data are so good as depicted in Tables 8-10, than I would suggest that they could use some kinetic modelling to allow them to predict the meat alteration at any combination of type of hydrolysate and concentration. It would significantly improve their manuscript and would be easier for the reader to follow the meat deterioration vs storage time.  

Author Response

Reviewer 1

Comments and Suggestions for Authors

This manuscript describes in detail a study of using protein hydrolysates as antimicrobails for meat shelf-life extension. In my opinion, the manuscript can be published with minor changes such as the following:

  • The title should be changed and include the biggest part of the study that is relevant to the characterization of the hydrolysates. Rephrase it so as to be easier to track the article and find information when searching for info relevant to the hydrolysates you studied.

Response:The title was rephrased accordingly

  • Table 5 needs to be improved. 

Response:Table 5 was improved and presented in landscape orientation for clean appearance

  • The results of Table 8, in my opinion does not significantly help the reader. Only 0 and 30 days results are presented and since the meat is studied and has a big variability in colour, I think that the authors should present the Total colour change (ΔΕ) index vs storage time. It would be easier and clearer for the reader to check whether there is a positive effect of using the hydrolysates and which one and at what concentration.

Response: The ΔΕ during storage period was calculated and converted to histogram.

  • In general, the authors did not model any of their results. If the data are so good as depicted in Tables 8-10, than I would suggest that they could use some kinetic modelling to allow them to predict the meat alteration at any combination of type of hydrolysate and concentration. It would significantly improve their manuscript and would be easier for the reader to follow the meat deterioration vs storage time.  

Response: The data in Tables 8-10 were predicted with trendline regression curves in Excel

Reviewer 2 Report

Comments and Suggestions for Authors

Manuscript ID: molecules-1279888

Title: Preserving and Enhancing Raw Chicken Meat by Bioactive Peptides Isolated from White, Red, and Black Kidney Beans Obtained by Enzymatic Hydrolysis

General comments:

In this manuscript, Saad et al. explored the potential effect of bioactive peptides from white, red and black kidney beans obtained by enzymatic hydrolysis on the preservation of raw chicken meat. The subject is interesting and topical. It is a very long manuscript, with a wide range of biochemical, physicochemical and microbiological methods. Overall, the manuscript is well written and well presented. However, a number of points need to be addressed before the manuscript is published.

First of all, the length of the manuscript as well as the number and diversity of experiments performed make the work difficult to follow from the beginning to the end. I suggest to the authors to reformulate the objectives by adding specific objectives in the introduction. For example, one of the specific objectives could be the protein and peptides isolation, another one the characterization of protein and hydrolysates, and a third one the Preservation of raw chicken meat. In this case the title of the manuscript should also be revised to fit the content adequately.

A second issue is that the authors quoted a considerable number of references in their work, 120 in total. This seems a bit too much and hence the question about the novelty of the manuscript. I would suggest that the authors reduce the number of references, focus on the most relevant ones, and make it clear at what level they are proposing stands out from the abundant literature on this topic.

The protein extraction was performed for two hours at room temperature. Did the authors think about the probable effect of intrinsic proteases which are generally present in such extracts and which could strongly impact the quality and composition of the extracted proteins? Wouldn't the use of inhibitor here have improved the extraction procedure? This can be seen on the SDS PAGE. As claimed by the authors, storage proteins (vicilin - 7S and phaseolin - 8S constitute the main proteins. By observing the SDS PAGE profile (Figure 1), it becomes clear that the protein bands obtained between 11-20 kDa are much more intense and represent the major part of the proteins. Is that not the result of an intrinsic proteolytic activity that takes place during the extraction process? The authors should clarify this.

Finally, knowing that hydrolysates contain not only bioactive peptides, but also several other compounds such as polyphenols, how can it be explained that the observed effects on meat preservation are mainly due to peptides?

Other remarks:

  1. Line 56 - define CDC. Same as line 60, please define BHA, and BHT. Abbreviations should always be defined when they first appear in the text.
  2. Line 65 : “The new trend in food preservation uses bioactive proteins as natural additives to preserve milk …” Is it here bioactive peptide or bioactive proteins?
  3. Line 88-89 : “The hydrolysis activity was more activity than trypsin and papain”, please check that sentence.
  4. Line 104: Phaseolus Vulgaris should be written in italics
  5. Lines 103-113 : The references of the suppliers are not complete - the name of the cities ? Same at line 117. Please also correct this throughout the text.
  6. Line 127: Why did the authors choose Alcalase as enzyme to perform the hydrolysis?
  7. Lines 128-129 : “The temperature was raised to 90ºC to inhibit the enzyme” for how long?
  8. Line 140-144: the SDS-PAGE was performed under reducing or non-reducing conditions? This paragraph should be described with more information about the sample preparation procedure
  9. Line 152 : Why use distilled water instead of buffer solutions?
  10. Line 153 : « Stirred »
  11. Line 270 : what is the meaning of « sanitize conditions » ?
  12. Lines 468-469 : “Enzymatic hydrolysis increased the total polyphenols by breaking down the protein–polyphenol complexes and releasing some polyphenols entrapped in the peptide fragments”. What type of enzymatic hydrolysis is involved here? Knowing very well the specificity of the enzymatic reaction, would Alcalase be able to hydrolyze the protein-polyphenol bonds?

Author Response

Reviewer 2

Comments and Suggestions for Authors

Manuscript ID: molecules-1279888

Title: Preserving and Enhancing Raw Chicken Meat by Bioactive Peptides Isolated from White, Red, and Black Kidney Beans Obtained by Enzymatic Hydrolysis

General comments:

In this manuscript, Saad et al. explored the potential effect of bioactive peptides from white, red and black kidney beans obtained by enzymatic hydrolysis on the preservation of raw chicken meat. The subject is interesting and topical. It is a very long manuscript, with a wide range of biochemical, physicochemical and microbiological methods. Overall, the manuscript is well written and well presented. However, a number of points need to be addressed before the manuscript is published.

Response: Thanks for the reviewer, and all needed points will be addressed

First of all, the length of the manuscript as well as the number and diversity of experiments performed make the work difficult to follow from the beginning to the end. I suggest to the authors to reformulate the objectives by adding specific objectives in the introduction. For example, one of the specific objectives could be the protein and peptides isolation, another one the characterization of protein and hydrolysates, and a third one the Preservation of raw chicken meat. In this case the title of the manuscript should also be revised to fit the content adequately.

Response: The objectives in introduction were reformulated accordingly; The rest of the manuscript followed the same order.

A second issue is that the authors quoted a considerable number of references in their work, 120 in total. This seems a bit too much and hence the question about the novelty of the manuscript. I would suggest that the authors reduce the number of references, focus on the most relevant ones, and make it clear at what level they are proposing stands out from the abundant literature on this topic.

Response: We reduced the number of cited references from 120 to 78. Then we were obliged to add the new references covering the suggested added part. So the final number of references is now 93. The cited references only explained the background of some current results but were not similar to them, so the novelty is still high.

The protein extraction was performed for two hours at room temperature. Did the authors think about the probable effect of intrinsic proteases which are generally present in such extracts and which could strongly impact the quality and composition of the extracted proteins?

Response: According to practical experience, intrinsic proteases are not expected to exert any considerable effect particularly within the short duration of (0-60 min) hydrolysis process. Any case we used control of protein without adding any external enzyme under the same conditions. We did not achieve any protein hydrolysis in this content

Wouldn't the use of inhibitor here have improved the extraction procedure? This can be seen on the SDS PAGE.

Response: Using some inhibitors may be useful. However, in this study our control was the protein without any proteolytic activity.Therefore,we could manage the proteolytic activity from the beginning by controlling the hydrolysis time without the need of enzyme inhibitors. We also tested the differenthydrolysates with different degree of hydrolysis to find a relation with the antioxidant and antimicrobial activity. Finally, we used the degree of hydrolysis which manifested the highest antioxidant and antimicrobial activity.

As claimed by the authors, storage proteins (vicilin - 7S and phaseolin - 8S constitute the main proteins. By observing the SDS PAGE profile (Figure 1), it becomes clear that the protein bands obtained between 11-20 kDa are much more intense and represent the major part of the proteins. Is that not the result of an intrinsic proteolytic activity that takes place during the extraction process? The authors should clarify this.

Response: We were referring to the degraded banded which disappeared in the hydrolysate as compared to the intact protein. We attributed the biological activity to the fainted or vanished bands as an indicator to the generation of active peptides. Intrinsic proteases are not expected to exert any considerable effect particularly within the short duration of (0-60 min) hydrolysis process. Therefore, we were more focused on the hydrolysis induced by the external enzymes, which we intended to investigate. Moreover, the intense bands 11-20 kda were attributed to polyphenol-protein complex that support the resistance of alcalse hydrolysis.

Finally, knowing that hydrolysates contain not only bioactive peptides, but also several other compounds such as polyphenols, how can it be explained that the observed effects on meat preservation are mainly due to peptides.

Response: In our control, we used the intact protein as mixed with the natural several compounds such as polyphenols. After hydrolysis the protein becomes degraded into small peptides while it is mixed with the same natural several compounds, e.g., polyphenols. Since there were considerable differences between the control and the hydrolysate and since they both are similar in their content of the natural compounds but different only in the protein status (intact or hydrolysed), then the activity should be attributed the protein status.Since the ration of free polyphenol in obtained peptides was 1:100, therefore the main preservation effect refer to peptides.

Other remarks:

  1. Line 56 - define CDC. Same as line 60, please define BHA, and BHT. Abbreviations should always be defined when they first appear in the text.

Response:Done accordingly

  1. Line 65 : “The new trend in food preservation uses bioactive proteins as natural additives to preserve milk …” Is it here bioactive peptide or bioactive proteins?

Response:Done accordingly

  1. Line 88-89 : “The hydrolysis activity was more activity than trypsin and papain”, please check that sentence.

Response:The sentence was checked accordingly

  1. Line 104: Phaseolus Vulgaris should be written in italics

Response:Done accordingly

  1. Lines 103-113 : The references of the suppliers are not complete - the name of the cities ? Same at line 117. Please also correct this throughout the text.

Response:Done accordingly

  1. Line 127: Why did the authors choose Alcalase as enzyme to perform the hydrolysis?\

  Response: Alcalase is a commercial enzyme preparation from Bacillus licheniformis, which consists primarily of subtilisin A; subtilisin A is an endopeptidase with broad actions, preferably cleaving terminal hydrophobic amino acids(Ellaiah et al, 2002). Alcalase-derived hydrolysates not only have higher antioxidant activities than those from other peptidases but also are more resistant to digestive enzymes (Sarmadi et al, 2010). Alcalase is known for its effectiveness towards plant proteins. Furthermore, it has a specific activity targeting the hydrophobic bonds between ArgorLys-Xreleasinghydrophobic and alkaline peptideas which are known for their antimicrobial activities. It is known for its doablespecific, whichguarantees the reproducibility of its results.

References

Ellaiah, P., Srinivasulu, B., &Adinarayana, K. (2002). A review on microbial alkaline proteases.

Sarmadi, B. H., & Ismail, A. (2010). Antioxidative peptides from food proteins: a review. Peptides31(10), 1949-1956.

  1. -Lines 128-129 : “The temperature was raised to 90ºC to inhibit the enzyme” for how long?

Response:The duration was added

  1. Line 140-144: the SDS-PAGE was performed under reducing or non-reducing conditions? This paragraph should be described with more information about the sample preparation procedure

Response:It was done in reducing conditions where loading buffer containing beta-    mercaptoethanol to reduce disulphide bridges in proteins. Moreover, separation method was detailed

  1. Line 152 : Why use distilled water instead of buffer solutions?

Response: Solubility measurements in buffer alone are difficult to reproduce, because gels or supersaturated solutions often form, making it impossible to determine solubility values for many proteins (Kramer et al, 2012). Measuring solubility in pure water is more meaningful for ordinary applicatios.

Kramer, R. M., Sh ende, V. R., Motl, N., Pace, C. N., &Scholtz, J. M. (2012). Toward a molecular understanding of protein solubility: increased negative surface charge correlates with increased solubility. Biophysical journal102(8), 1907-1915.‏

  1. Line 153 : « Stirred »

Response:Done accordingly

  1. Line 270 : what is the meaning of « sanitize conditions » ?

Response:It was cleared accordingly

  1. Lines 468-469 : “Enzymatic hydrolysis increased the total polyphenols by breaking down the protein–polyphenol complexes and releasing some polyphenols entrapped in the peptide fragments”. What type of enzymatic hydrolysis is involved here?

Response:It was pepsin enzymatic hydrolysisand the paragraph was reformulated to be clear.

Knowing very well the specificity of the enzymatic reaction, would Alcalase be able to hydrolyze the protein-polyphenol bonds?

Response: Actually, the enzyme hydrolyze the protienwhich bound with polyphenols.Therefore, the polyphenols will be released and their free content will increase (Torre et al., 2008). Additionally, it was found that alcalase hydrolysis of rice bran protein for 10 min was enough to extract all the bound phenolic acids (Vijitpunyaruk, and Theerakulkait, 2014).

References:

Vijitpunyaruk, T., &Theerakulkait, C. (2014). Preparation of alcalasehydrolysed rice bran protein concentrate and its inhibitory effect on soybean lipoxygenase activity. International Journal of Food Science & Technology49(2), 501-507.‏

Torre, P., Aliakbarian, B., Rivas, B., Domínguez, J. M., &Converti, A. (2008). Release of ferulic acid from corn cobs by alkaline hydrolysis. Biochemical Engineering Journal40(3), 500-506.‏

Round 2

Reviewer 2 Report

Good job